# Catalytic Direct Decomposition of NO$_x$ Using Non-Noble Metal Catalysts

**M. K. Shukla [1], Balendra V. S. Chauhan [2] , Sneha Verma [3],* and Atul Dhar [4],***

[1] Automotive Fuels and Lubricants Application Division, CSIR-Indian Institute of Petroleum, Dehradun 248005, Uttarakhand, India

[2] Centre for Earth Observation Science, School of Applied Sciences, University of Brighton, Brighton BN2 4GJ, UK

[3] School of Science and Technology, City University of London, London EC1V 0HB, UK

[4] School of Engineering, Indian Institute of Technology, Mandi 175005, Himachal Pradesh, India

* Correspondence: sneha.verma@city.ac.uk (S.V.); add@iitmandi.ac.in (A.D.)

**Abstract:** Nitrogen oxides (NO$_x$) gases, such as nitrous oxide (N$_2$O), nitrogen oxide (NO), and nitrogen dioxide (NO$_2$), are considered the most hazardous exhausts exhaled by industries and stationary and non-stationary application engines. Investigation of catalytic decomposition of NO has been carried out on copper ion exchanged with different bases, such as COK12, Nb$_2$O$_5$, Y-zeolite, and ZSM5. The catalytic decomposition of NO is widely accepted as an excellent method for the abatement of NO. However, the catalyst that achieves the highest reactivity in terms of NO decomposition is still a matter of research. The present paper aims to extend the research on the reactivity of non-noble metal-based catalysts using the direct decomposition method to remove NO from diesel engine exhaust. The reactivity of catalysts was observed in a quartz fixed bed reactor of 10 mm diameter placed in a furnace maintained at a temperature of 200 °C to 600 °C. The flow of NO was controlled by a mass flow controller, and the gas chromatography technique was used to observe the reactivity of the catalysts. Analysis showed that adding Cu to COK12, Nb$_2$O$_5$, Y-zeolite, and ZSM5 supports resulted in a rise in NO decomposition compared to stand-alone supports. Further experimental trials on the performance of Cu-ZSM5 at varying flow rates of NO showed that the NO decomposition activity of the catalyst was higher at lower flow rates of NO.

**Keywords:** abatement; decomposition; gas chromatography; NO$_x$; non-noble metals





## 1. Introduction

The increasing global population has caused a consequent increase in energy demand, resulting in higher employment of natural resources and causing an increase in environmental pollution. Nowadays, the environment is full of automotive and industrial exhaust and power plants' harmful emissions. In the atmospheric pollution caused by industries and vehicle emissions, nitrogen oxides are considered the primary pollutants compared to their counterparts, such as sulfur oxides, unburned hydrocarbons, and carbon monoxide.

NO$_x$ is the combination of two brutal pollutants, i.e., NO and NO$_2$ [1,2]. NO$_x$ is responsible for pressing problems, such as ozone layer depletion, acid rain, photochemical smog, tropospheric ozone, and global warming [3,4]. The two primary sources of NO$_x$ are industries and the transport sector [5]. Since the transport sector is widespread and uses small-bore and large-bore engine vehicles, it can be considered a significant source of NO$_x$ emissions [6,7]. However, vehicles also emit such pollutants as CO, HC, soot, and PM. Among all other pollutants, NO$_x$ is believed to be the most harmful. In addition, vehicle manufacturers know very well that it is very tough to reduce these pollutants, even with technological advancements. Cellek M.S. [8] noted that in combustion, there are three types of NO$_x$: firstly "fuel NO$_x$", in which fuels containing nitrogen are oxidized, resulting in NO$_x$ emission; secondly, "prompt NO$_x$," which is formed when nitrogen in

the air reacts with fuel during fuel-rich conditions; and thirdly, "thermal $NO_x$," in which the concentration of $NO_x$ depends on three factors—molar concentration of oxygen, molar concentration of nitrogen, and combustion temperature. The primary form of $NO_x$ emission is NO, produced during combustion. Complex $NO_x$ formation can be easily understood by the Zeldovich mechanism (Equations (1)–(3)). This mechanism states that NO is generated when the available oxygen limit is ~200,000 ppm and the temperature is greater than 1300 °C.

$$N_2 + O \rightarrow NO + N \tag{1}$$

$$N + O_2 \rightarrow NO + O \tag{2}$$

$$N + OH \rightarrow NO + H \tag{3}$$

$NO_x$ does not come from normal conditions. It forms in significant amounts once the flame temperature reaches a threshold of 2800 F (~1537 °C) [9]. $NO_x$ formation is highest when the air: fuel ratio in combustion attains 5–7% of excess $O_2$ (25–45% excess air) [9]. Low-excess air causes oxygen deficiency in reaction, and high-access air drives down the flame's temperature by slowing down the reaction rate.

$NO_x$ emissions have an adverse effect on human health directly and indirectly. Instantly, they create headaches, breathing problems, and eye irritation and corrode the teeth. Indirectly, $NO_x$ damages the water, the atmosphere, and land. Therefore, regulatory bodies prepared stricter emission norms, globally known as Euro norms, and different countries formulated theirs in compliance with these norms. India is not behind in this regard, and has set up Bharat stage (BS) norms for petrol and diesel engines.

$NO_x$ can be reduced via two strategies: (i) creating/maintaining proper chemical reaction conditions and (ii) reaction of tail-out $NO_x$ emissions with other reagents and breaking these down into less harmful pollutants. However, controlling a chemical reaction during combustion is quite tricky; thus, the second approach is generally used in the automotive industry. The second approach can be achieved in four ways. These are selective catalytic reduction (SCR), nonselective catalytic reduction (NSCR), $NO_x$ storage/reduction (NSR), and NO direct decomposition. SCR technology was developed in the 19th century to reduce the $NO_x$ in power-generation plants, using ammonia ($NH_3$) as a reagent. It was first accepted and introduced in the 1970s in Japan. Germany adopted this technology to curb $NO_x$ in 1985 [10]. The automotive industry has also adopted this technology in internal combustion engines (ICEs) used for transport applications and off-road machinery. SCR is an advanced emission technology where liquid reductant agent is introduced through a particular stream. Generally, the reductant agents used are diesel exhaust fluid (DEF) and automotive/industry grade materials. In this complex system, ammonia is used as an inherent reductant contained in the system. This system can reduce $NO_x$ by 90%. However, during adaptations of this technology, challenges were experienced by researchers, such as space limitations in the engine system, less conversion efficiency, a narrow temperature window, availability of ammonia, and safe handling of ammonia. NSR ($NO_x$ storage reduction) technology converts $NO_x$ and $N_2$. First, exhaust gas is routed to $NO_x$ storage materials using PGM and then purified into $N_2$ gas. This is mainly used for highly efficient diesel engines. Additionally, NSCR is a technology where $NO_x$, CO, and HCs are converted into $N_2$ and $CO_2$ with the help of a catalyst. The most important part of this technique is that it does not require any reagent, because HCs work as a reductant. NSCR is generally used as a three-way catalyst in the automotive industry. The efficiency reported/realized during testing for $NO_x$ removal is 90–98% [11]. Among all other routes, direct decomposition of NO is the best to curb overall $NO_x$ emissions.

This method is thermodynamically favorable, and no extra reductants are required. In the literature, simple metal oxides, complex metal oxides, noble metals, rare-earth oxides [12,13], perovskite-type oxides [14,15], $Co_3O_4$ [16], and zeolites [16,17] have been reported to have potential as active catalysts for various applications. The rare-earth oxides of type C are the most favorable, because of their high tolerability in the presence of $CO_2$ and $O_2$ and the availability of ample cavity space for NO decomposition. For the first

time in history, Jellinek [18] tried to decompose nitric oxide by the streaming method. However, the theoretical background was not established, and it was difficult to conclude whether the reactions were homogeneous or partly heterogeneous. Based on his findings, Green and Hinshelwood [17] tried to decompose nitric oxide on a platinum surface. They also observed a reaction rate on the heated platinum surface over a different temperature range, where maximum and minimum decomposition were 13% and 60%, respectively, subject to the exposure time. Shelef et al. [19] investigated the NO decomposition rate using a UOP catalyst and cobalt oxide on an alumina-silica support. They observed a decomposition rate in the range of 30–100% for the UOP catalyst and 6–35% for cobalt oxide on the alumina-silica support. Masuiet et al. [12] investigated C-type cubic $Y_2O_3$-$Tb_4O_7$-$ZrO_2$ for direct decomposition of NO. They found peak NO to $N_2$ conversion of ~67% at 900 °C using a $(Y_{0.67} Tb_{0.30} Zr_{0.03})2O_{3.33}$ catalyst in an atmosphere of NO/He. However, with identical catalysts, they observed an even higher conversion rate in $O_2/CO_2$ atmosphere. They recommended the use of $Y_2O_3$-$Tb_4O_7$-$ZrO_2$ for NO decomposition as a catalyst. Tsujimoto et al. [20] used a C-type cubic holmium oxide ($Ho_2O_3$) catalyst to introduce praseodymium and zirconium ions into a lattice for increasing the surface area. They found a maximum 71% NO-to-$N_2$ conversion rate in the absence of NO/He atmosphere at 900 °C. This desirable conversion rate was not limited to only a lack of NO/He atmosphere: they also found 50% and 47% conversion rates in the presence of 5% vol $O_2$ and $CO_2$, respectively. Tsujimoto et al. [20] investigated the influence of Tb and Ba on C-type cubic rare earth oxides based on $Y_2O_3$ for direct NO decomposition reaction. They concluded that the addition/doping of $Ba^{2+}$ influenced two critical parameters in an incremental way: (i) oxide anion vacancies in the lattice and (ii) primary sites. With the use of $Ba^{2+}$, they found $Tb^{3+/4+}$ to be the best doping material, with enhanced catalytic activity. However, in other studies, they recommended the use of $(Y_{0.89}Zr_{0.07}Ba_{0.04})2O_{3.03}$ [21], $(Gd_{0.70}Y_{0.26}Ba_{0.04})2O_{2.96}$ [22], and $(Y_{0.97}Zr_{0.03})2O_{3.03}$ [21] over simple C-type cubic rare-earth oxides based on their high activity rates. Doi et al. [23] examined direct decomposition of NO on Ba catalysts experimentally, where rare-earth oxides were used as supports. They concluded that since rare-earth oxide catalyzed NO decomposition reactions effectively, the doping of Ba, Ba-$Y_2O_3$, Ba-$Dy_2O$, and Ba-$Sm_2O_3$ increased catalytic activity markedly. Imanaka et al. [21,22,24–27] recommended using rare-earth oxide (REO)-based catalysts, and concluded that the C-type cubic structure was the primary factor responsible for high NO decomposition activity. Vannice et al. [28] and Xi et al. [29,30] also recommended using alkaline earth oxides, i.e., Ba/MgO and Sr/$La_2O_3$, which unfortunately do not possess inherent oxygen-deficient sites. For these oxides, they reported $Y_2O_3$ to be the best practical support and found NO decomposition activity in a decreasing order of Ba/$Y_2O_3$ > Sr/$Y_2O_3$ > Ca/$Y_2O_3$ > Mg/$Y_2O_3$ > $Y_2O_3$ [23]. Other than this, several rare-earth oxides have tested, with NO conversion rates found from 0 to 100%, depending on the temperature [16]. In addition, various perovskite-type and perovskite-related oxides have tested, and the NO conversion rate was found to be 0–90%; however, it depended on the temperature [16]. From the literature, important potential routes to eliminate $NO_x$ are (i) use of SCR in the presence of $NH_3$ and (ii) NO decomposition in $N_2$ and $O_2$. However, SCR has an inherent problem with the usage of $NH_3$. Another way is emerging as the best way to eliminate $NO_x$ emissions is a reducing agent, and during decomposition, no harmful products are formed. However, low conversion efficiency below 800 °C temperature and high cost are researchers' central issues. The primary responsible factor for the high cost is noble metals, i.e., Pd, Pt, and Rh, so, researchers have moved to find a low-cost solution. In recent studies, Cu-ZSM5 zeolites and Cu-based catalysts have gained attention as they are relatively cheaper than noble metals. Matsumoto et al. [31] studied the influence of Cu-$ZSM_5$ catalyst on $NO_x$ conversion. They mounted the Cu-ZSM5 catalyst in the exhaust system and found that $NO_x$ conversion was significant at 300 °C temperature and found the maximum conversion as 40%. However, at 400 °C, $NO_x$ conversion was dropped significantly. This investigation underscores that the temperature window should be expanded to exploit the advantage of Cu-ZSM5 for $NO_x$ conversion. As Cu is a cost-

effective active catalyst, Komvokis et al. [32] tried Cu-exchanged ZSM5 zeolites with coated $CeO_2$ nanoparticles for $NO_x$ conversion. Their results were compared with the non-coated Cu-ZSM5 catalyst. They reported low maximum NO conversion activity in comparison with non-coated Cu-ZSM5 catalyst. They recommended the use of coated Cu-ZSM5 catalyst at low temperatures (350 °C) as reactivity was comparable to the non-coated Cu-ZSM5. However, in the literature, few drawbacks have been observed/reported during the extensive testing of catalyst. Cu-ZSM5 catalyst has poor activity in the presence of water, due to structural change/collapse of and redistribution of $Cu^{2+}$ ions over a zeolite catalyst [33]. Dealumination of the zeolite occurs over long-term exposure to the wet conditions, which negatively affect the $Cu_{2+}$ exchange sites. Since the advantage of Cu usage is relatively high, it was also used in other applications where high reaction activity is expected. Hence, the present paper is on the Cu metal-based catalyst's activity towards the NO decomposition.

The aim of the present investigation was to examine low conversion efficiency and reduce the overall cost for emission prevention. Different supports were selected for the study to encounter the first route, i.e., COK12, Y-Zeolite, ZSM5, and $Nb_2O_5$. Copper was chosen instead of noble metals to reduce the cost of this entire process. This study establishes the possibility of this route to reduce $NO_x$ emissions effectively, and this paper will further discuss the NO direct decomposition method.

## 2. Materials and Methods

### 2.1. Materials

Y-Zeolite, ZSM5, were obtained from Zeolyst and $Nb_2O_5.xH_2O$ from CBMM, Brazil. Copper Nitrate Trihydrate ($Cu(NO_3)_2 \cdot 3H_2O$) and Pluronic P123 were obtained from Sigma Aldrich. Sodium silicate solution (10% NaOH, 27% $SiO_2$) and nitric acid were obtained from Merck, Germany. COK12 is prepared in the laboratory of CSIR-Indian Institute of Petroleum, Dehradun.

### 2.2. Catalyst Preparation

The wet impregnation method was used to prepare the catalysts. The commercial Y-Zeolite, ZSM5, and $Nb_2O_5.xH2O$ were purchased from Zeolyst. Before impregnating Cu, the hydrated supports (Y-Zeolite, ZSM5, $Nb_2O_5.xH_2O$) were treated at 500–600 °C for 5–8 h to remove the water/moisture. The self-assembly method using surfactant P123 was employed for the processing of COK12 (mesoporous silica) support. In this method, solution A was prepared by dissolving 3.65 g monohydrate citric acid, 4.05 g pluronic acid, and the 2.62 g dehydrated trisodium citrate in the 107.55 g of distilled water; the solution is then stirred for at least 24 h at room temperature. Then, solution B is prepared by dissolving the solution of 10.4 g of sodium silicate (10% NaOH, 27% $SiO_2$, Merck, Darmstadt, Germany), and further, this solution was diluted with 30.0 g of $H2O$. In the next step, the obtained solution B was added to solution A. With the help of the magnetic stirrer, the resulting mixture was kept in the stirrer for 5 min at 175 rpm, and then it was kept untouched for 24 h. The material obtained was filtered, washed, and dried at 60 °C overnight. The calculations of dry materials were performed in two steps, first for 8 h at 300 °C and then for 8 h at 500 °C with ramp 1 °C/min. For metal, impregnation required amount of metal salt $Cu(NO_3)2 \cdot 3H_2O$ (Sigma Aldrich, >98%) was dissolved in distilled water. The pre-treated support was then added to the metal salt solution under constant mechanical agitation at 80 °C until the water gets evaporated. The obtained solid was calcined at 600 °C for 6 h in a muffle furnace to obtain a catalyst. The rationale for the addition of 3% Cu is based on the literature, ZSM5 is being used by Curtin et al. [34] they did experimental investigations for direct decomposition of NO using Cu-ZSM5 catalyst with varying weight percentages of Cu in the catalyst, they found that for weight percentages of Cu from 2.5% to 3% the NO conversion was optimum and as maximum (65%).

### 2.3. Experimental Setup

A fixed bed quartz glass reactor was used for the reactivity tests of catalysts activity towards NO decomposition. Jia et al. also used a fixed bed reactor setup for the removal of $Hg^0$ from the modified biochar, later did a temperature programmed desorption analysis [35]. The quartz glass reactor idea is followed from the studies of S.B. Jorgensen [36] and Stegehake et al. [37] where the modelling and validation of fixed bed reactors, and their applicability for reaction setup is praised. The prepared setup was equipped with a quartz reactor, furnace, thermocouples, PID controllers, mass flow controllers (MFCs), gas regulators, gas shut ON/OFF valves, and temperature control units. The Figure 1a shows the schematic of the experimental setup and Figure 1b gives the labelled visualization of the setup. Nitrogen gas was used as a purging agent to remove any unwanted gases present in the system. After the purging, the fixed bed reactor was loaded with the prepared catalyst, and NO gas was made to pass through the reactor. For our case, the flow rate of NO gas was varied (with the help of MFC) as 20 mL/min, 50 mL/min, or 100 mL/min. Further, a temperature control unit controlled the reactor's temperature (as this fixed bed reactor was vertically placed in the furnace which can be seen in the experimental setup).

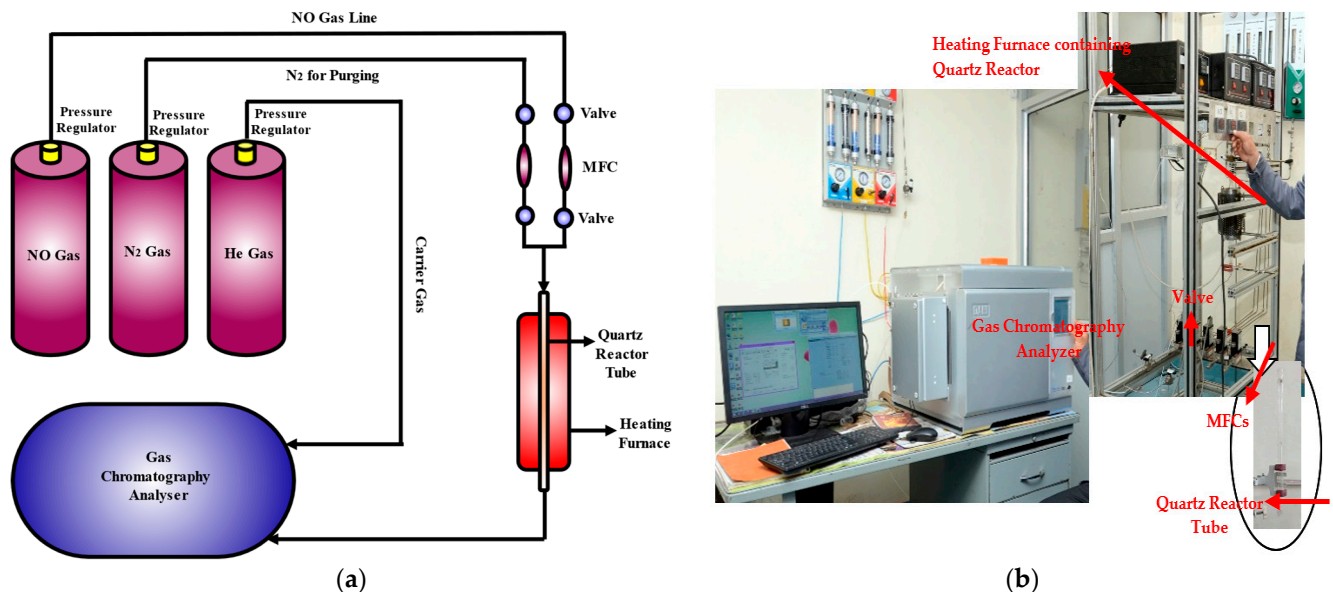

(**a**)        (**b**)

**Figure 1.** (**a**) Schematic of the experimental setup, (**b**) Experimental setup used for NO conversion investigations.

For the present study, the temperature was varied from 200 °C to 600 °C. Blank tests were also performed over quartz wool beds. Further, the NO gas passed through the fixed bed reactor setup was taken into a Gas Chromatography machine (model: DANI Master GC) equipped with the thermal conductivity detectors, which was used to measure the NO concentration change after the gas is passed through the reactor.

### 2.4. Characterization Methods

The prepared catalysts' surface areas have been measured by nitrogen adsorption at 77 K using the Brunauer–Emmett–Teller (BET) method and Barrett, Joyner, and Halenda (BJH) method and was adopted to find the pore size from the desorption isotherm. Before the analysis, the sample was outgassed at 523 K for 4 h. The surface area measurements have been performed on the Micromeritics 3-Flex Physisorption instrument. Crystalline phases were studied using the Proto-manufacturing AXRD benchtop powder diffraction system from $2\theta = 2°$ to $80°$. The resultant peaks observed were compared to the standard JCPDS data. The XRD patterns were recorded with a step size of $0.04°$. TEM was conducted on the JEOL JEM-2100 analytical electron microscope operating at 200 kV equipped with

a LaB6 crystal as an electron source. The morphology of the catalyst was analyzed using SEM analysis. An FEI Quanta 200 F (FEI Company, Eindhoven, The Netherlands) SEM instrument has been used to perform the SEM analysis. The redox behavior and the oxygen storage capacity of the catalysts were analyzed by performing temperature-programmed reduction experiments. The Micromeritics Auto Chem II apparatus was used to carry out the temperature-programmed reduction experiments. The sample was placed in the TPR cell and flushed with argon for 30 min at 150 °C. The sample was then subsequently cooled down to room temperature. Finally, the furnace temperature was increased at a ramp-up rate of 10 °C/min in a 15 mL/min flow rate with the H2/Ar mixture (10:90 ratios). A thermal conductivity detector (TCD) was used to monitor the signals corresponding to H2 consumption. $CO_2$-TPD and $NH_3$-TPD measurements have also been completed for the prepared catalysts. TPD studies were carried out using an Autochem HP II-2950 chemisorption analyzer (Micromeritics, Norcross, GA, USA). NH3-TPD was performed on Micromeritics, Autochem HP II-2950, equipped with a TCD. Before analysis, the sample was pre-treated at 300 °C for 1 h in He, and then it was exposed to 30 mL·min$^{-1}$ flow of 10% NH3-He for 30 min. After the adsorption, it was exposed to He for 30 min to remove extra NH3-He present over the surface; then, the temperature was raised to 700 °C for TPD measurements.

## 3. Result and Discussion

### 3.1. Material Characteristics

3.1.1. Surface Area

The textural properties of Cu-COK12, Cu-Y-Zeolite, ZSM5, Cu-ZSM5, and Cu-Nb$_2$O$_5$ catalysts were identified by N$_2$-adsorption/desorption analysis, and the results are displayed in Table 1.

**Table 1.** Surface Area measurement of all the catalysts used in the research.

| Catalyst | Surface Area, m$^2$/g | Pore Volume, cm$^3$/g | Pore Size, nm |
|:---:|:---:|:---:|:---:|
| COK12 | 270 | 0.79 | 11.80 |
| 3% Cu-COK12 | 323 | 0.72 | 8.90 |
| Nb$_2$O$_5$ | 15.4 | 0.06 | 16.00 |
| 3% Cu-Nb$_2$O$_5$ | 6.5 | 0.08 | 53.6 |
| Y-Zeolite | 325 | 0.47 | 7.45 |
| 3% Cu-Y-Zeolite | 230 | 0.50 | 8.70 |
| ZSM5 | 293 | 0.20 | 5.90 |
| 3% Cu-ZSM5 | 228 | 0.37 | 6.50 |

It is observed that the surface area is increased by 20% in the case of COK12 when modified with Cu. The pore size & pore volume was also decreased by 24% which suggests that the Cu species couldn't penetrate to the pores of COK12 but adsorbed over the COK12 surface. Whereas, in the case of Cu-modified Nb$_2$O$_5$, Y-Zeolite & ZSM5 the pore volume & pore size is increased, and the surface area decreased in all the Cu-modified catalysts. The pore volume & pore size is increased by 85% & 32% respectively in Cu-ZSM5 compared to ZSM5 which indicates a good level of penetration to the pores of ZSM5 by Cu species is occurred by changing the channels structure of ZSM5. In the case of Cu-modified Y-Zeolite & Nb$_2$O$_5$, the pore size increased by 50% & 235% respectively, but the pore volume is increased by 6% & 33% only. It indicated that the pores of Y-Zeolite & Nb$_2$O$_5$ are almost covered when modifying with Cu, but in the case of Cu-ZSM5, Cu is not covering the pores of ZSM5 & thus keeping the pores available for the reaction to happen. It may be attributed to the fact that after entering the pores, the Cu might break the Si-Al network in the channels of ZSM5 & formed its bond within ZSM5 network, making the larger pore

volume & pore size. Due to this, the reactant will have to react first with the Si-Al layer & then with the Cu particle. It will avoid the increment of Cu particle size during reaction & thus agglomeration & sintering. Although the surface area of ZSM5 is decreased with the addition of Cu, but a good conversion & a larger pore volume is worth considering when compared with the decrease in surface area from 293 $m^2/g$ to 228 $m^2/g$, the addition of Cu with ZSM5 resulted in a decrease in surface area; this can also be due to the fact that CuO particles (generally the copper species) may have partially covered the pores of the ZSM5's external surface or the channels. The same trend was observed in the case of Cu-modified Y-Zeolite, $Nb_2O_5$, and COK12 catalysts.

### 3.1.2. X-ray Diffraction

Figure 2 shows the XRD patterns of Cu-modified ZSM5, Y-Zeolite, $Nb_2O_5$, and COK12 catalysts with varying diffracted angles (2θ) against the intensity (a.u.). All typical peaks were compared with the XRD range of reported ZSM5, Y-Zeolite, $Nb_2O_5$, and COK12 catalyst. The Cu-ZSM5 catalysts' diffraction peaks were seen at 8.9°, 22.9°, 23.8°, and 24.3°, which are corresponding to Miller indices of ZSM5 support. Similarly in case of Cu-Y-Zeolite, typical peaks were observed at around 43° (112) and 68° (311) of Y-Zeolite. Further Cu-$Nb_2O_5$ catalyst shows $Nb_2O_5$ peaks designated as (111), (020), (111), (112), (222), (133), (321), (113), and (312) with an approximate diffraction angle of 23°, 29°, 37°, 47°, 51°, 56°, 58°, 64°, and 72°, respectively.

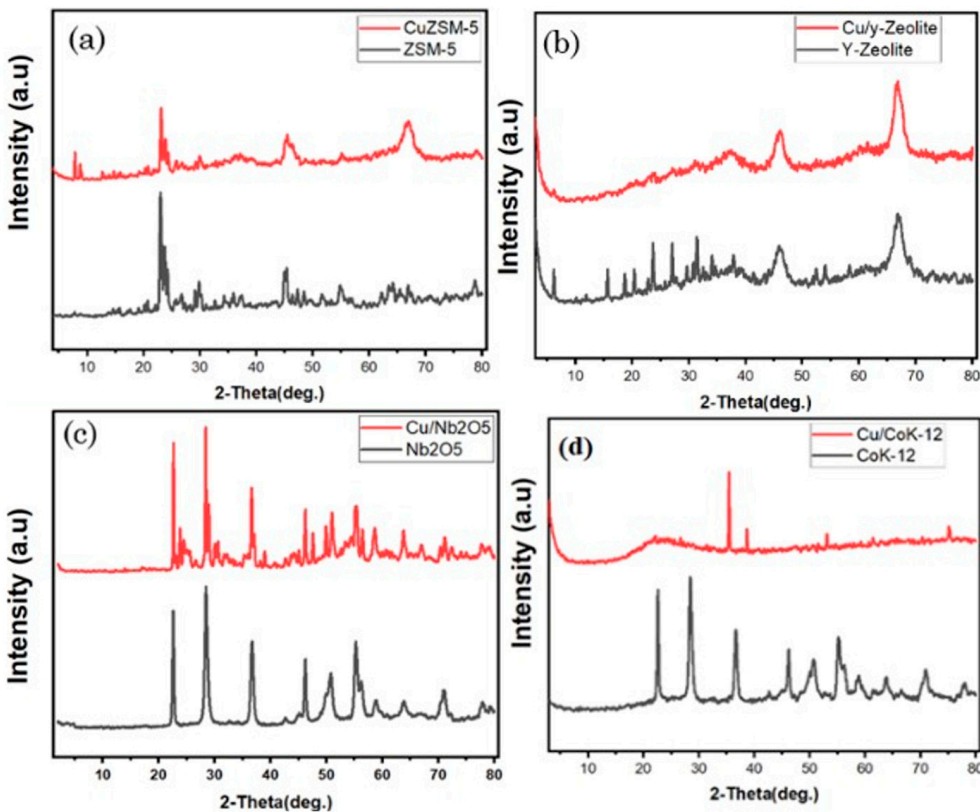

**Figure 2.** XRD pattern of (**a**) Cu-ZSM5 and ZSM5, (**b**) Cu-Y-Zeolite and Y-Zeolite, (**c**) Cu-$Nb_2O_5$ and $Nb_2O_5$, (**d**) Cu-COK12 and COK12.

This indicates that good dispersion of CuO and CuO impregnation does not do any structural damage in these supports [38]. However, the peak intensity of supports is lowering suggesting the impact of CuO particles on the pores of these supports [39] (Table 1). Only, Cu-COK12 catalyst shows four characteristics peaks of CuO at 35°, 38°, 54°, and 76°, indexed to (002), (111), (020), and (222) miller indices due to large CuO particles present on COK12 surface [40].

### 3.1.3. Transmission Electron Microscopy (TEM)

High Resolution TEM images of COK12 and Cu-COK12 catalyst are shown in Figure 3 at different magnifications (100 nm, 50 nm, and 20 nm). Figure 3a–c presents the TEM images of the parent COK12 catalyst, showing that the catalyst particles have honeybee comb-like morphology. Figure 3c indicates that the COK12 catalyst has more agglomerated particles with uniform shape and size. Figure 3d–f shows the Cu doped COK12 catalyst TEM images and confirmed that the particles of a doped sample are highly porous with a pseudo spherical shape. Also, Figure 3f at high magnification for the Cu-COK12 sample exhibits small black dots, clearly indicating the presence/impregnation of Cu on the surface of the support.

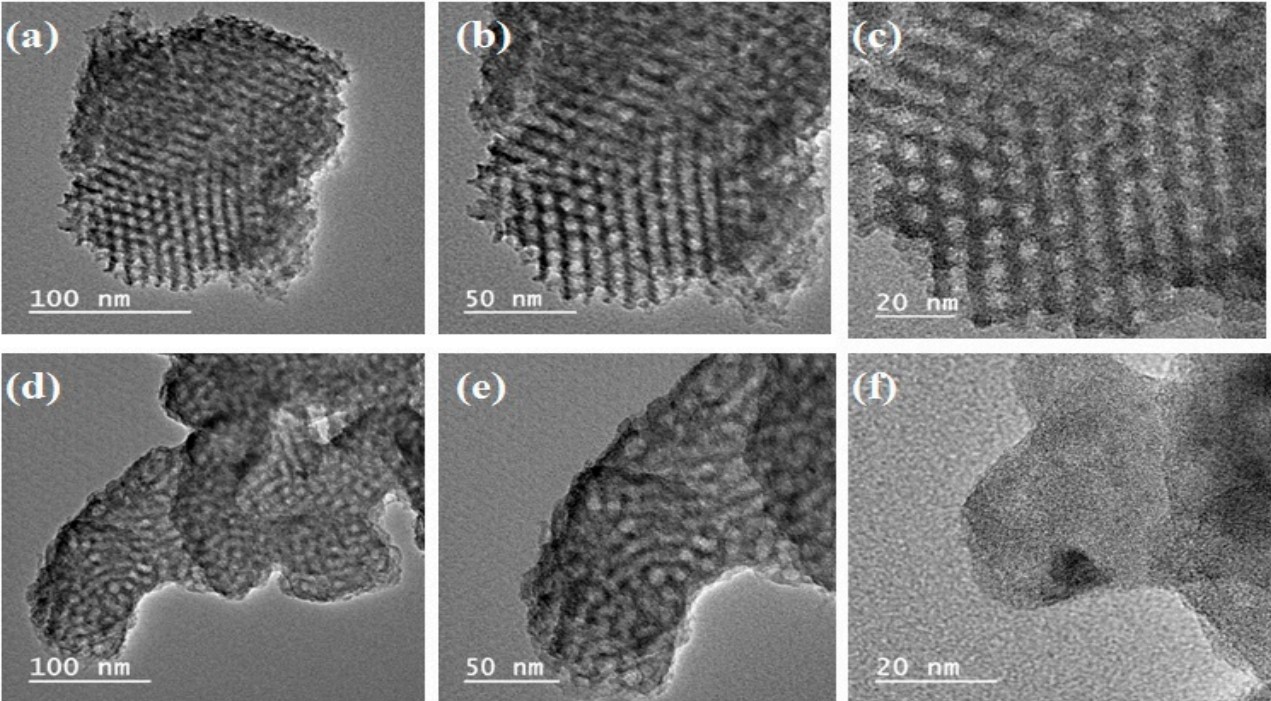

**Figure 3.** TEM images of COK12 (**a**–**c**) and Cu/COK12 (**d**–**f**).

The TEM images of $Nb_2O_5$ and $Cu-Nb_2O_5$ catalyst are shown in Figure 4 at different magnifications (100, 50, 20 nm). Figure 4a–c depicts the TEM image of the $Nb_2O_5$ catalyst, which shows the formation of the nano chain-like structure of particles of $Nb_2O_5$ with apparent aggregation. Moreover, Figure 4c at high magnification for $Nb_2O_5$ catalyst shows that particles are arranged in nano chain form with head and tail. Figure 4d–f displays the Cu doped $Nb_2O_5$ catalyst TEM images, which confirm that particles of $Cu-Nb_2O_5$ have a spherical shape with apparent aggregation non-uniform size. Figure 4f displays a high magnification view of the core shell like the $Cu-Nb_2O_5$ Catalyst structure. The TEM images of Y-Zeolite and Cu-Y-Zeolite catalyst are shown in Figure 5 at different magnifications (100, 50, 20 nm). All the images (Figure 5a–f) show the formation of flaky or elongated shape particles with agglomeration. In Figure 5c,f, a high magnification view of Y-Zeolite and Cu-Y-Zeolite indicates the catalyst particles' shape.

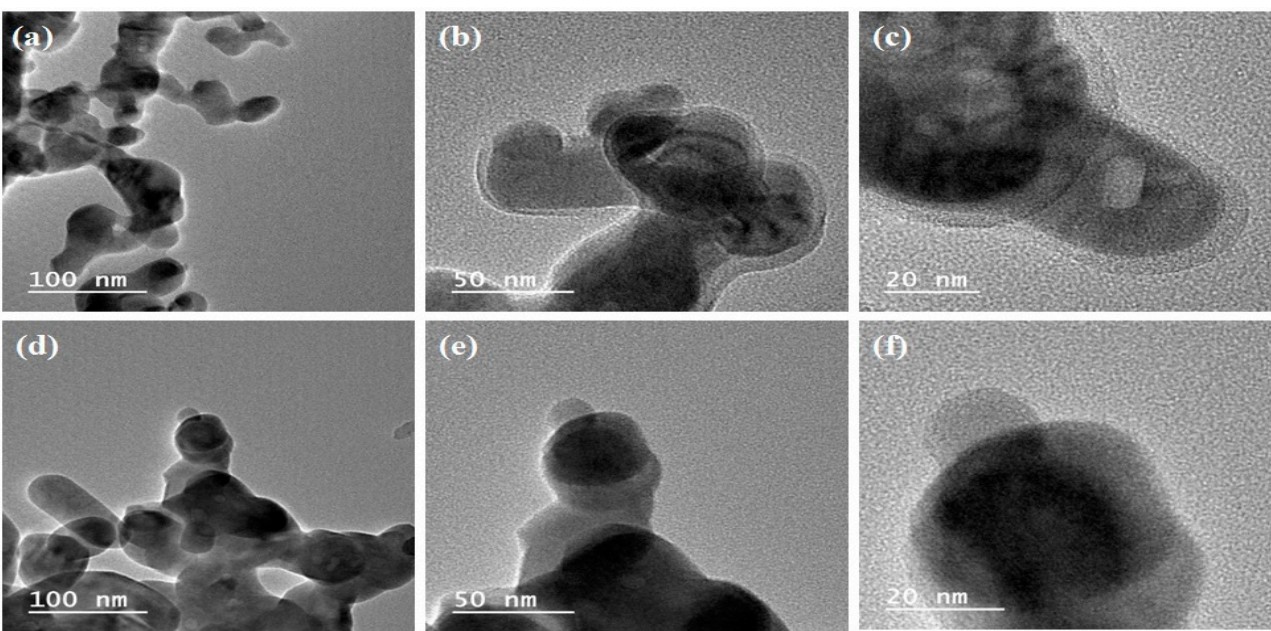

**Figure 4.** TEM images of Nb$_2$O$_5$ (**a**–**c**) and Cu-Nb$_2$O$_5$ (**d**–**f**).

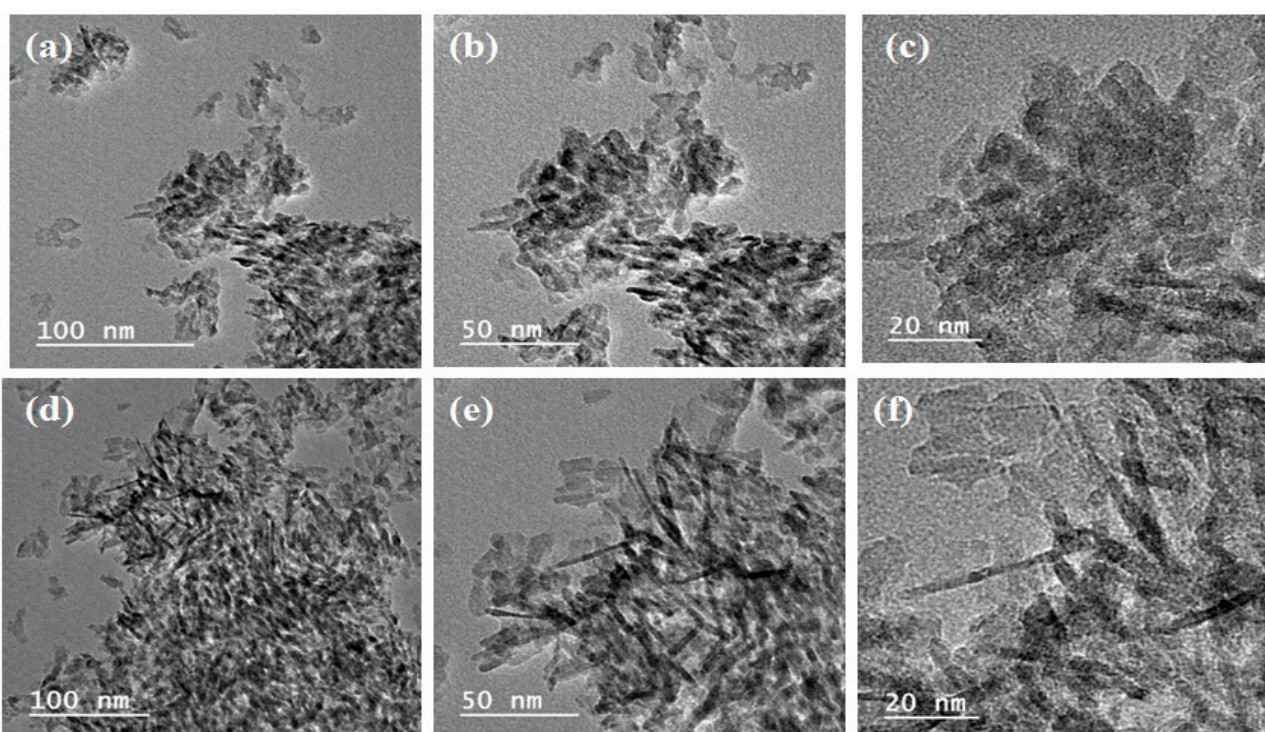

**Figure 5.** TEM images of Y-Zeolite (**a**–**c**) and Cu-Y-Zeolite (**d**–**f**).

The TEM images of the Cu-ZSM5 catalyst is displayed in Figure 6a–c. These images show that particles of catalyst Cu-ZSM5 are angular in shape with high agglomeration. Also, image (Figure 6c) with a highly magnified view of the catalyst shows the formation of angular shape particles arrange in net-like structure morphology.

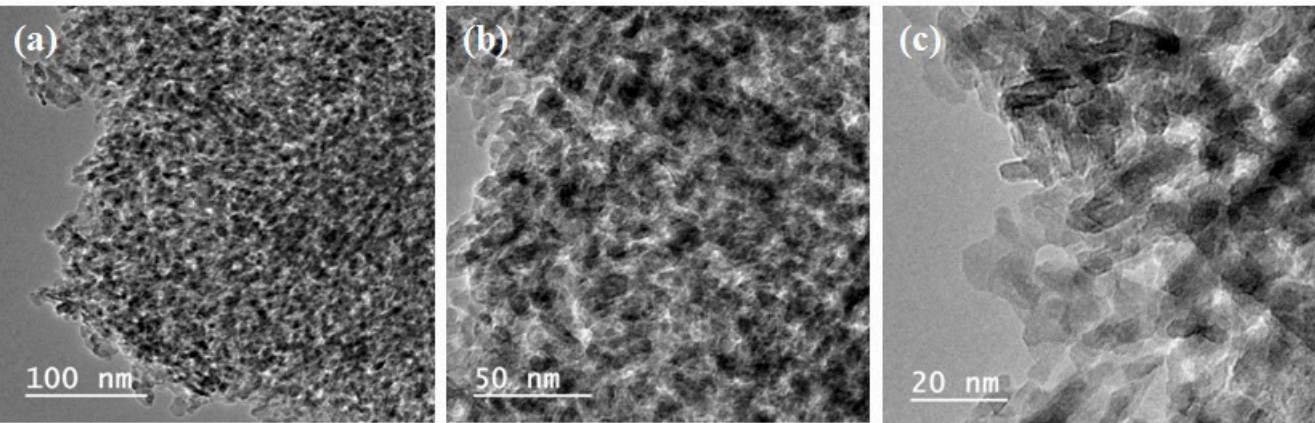

**Figure 6.** TEM images of Cu-ZSM5 (**a**–**c**).

### 3.1.4. Scanning Electron Microscopy (SEM)

Figure 7 shows the SEM images used to examine the surface morphology of Cu-modified ZSM5, Y Zeolite, $Nb_2O_5$, and COK12 catalysts. These images bring out the variation in the morphology as a function of the catalyst. Figure 7a tells that the Cu-COK12 catalyst has a cloudy morphology with highly agglomerated particles, and the particles have a non-uniform size with a smooth surface. The SEM images of the Cu-$Nb_2O_5$ catalyst are exhibited in Figure 7b, which indicates that the particles have a puros structure with a pseudo spherical shape. It is also observed that the particles are less aggregated than the Cu-COK12 catalyst. Figure 7c,d presents the morphology of Cu-Y-Zeolite and Cu-ZSM5 samples and indicates that the particles are high agglomerated with non-uniform shapes and sizes. Both images also confirm that the morphology is nonporous with a ruff surface.

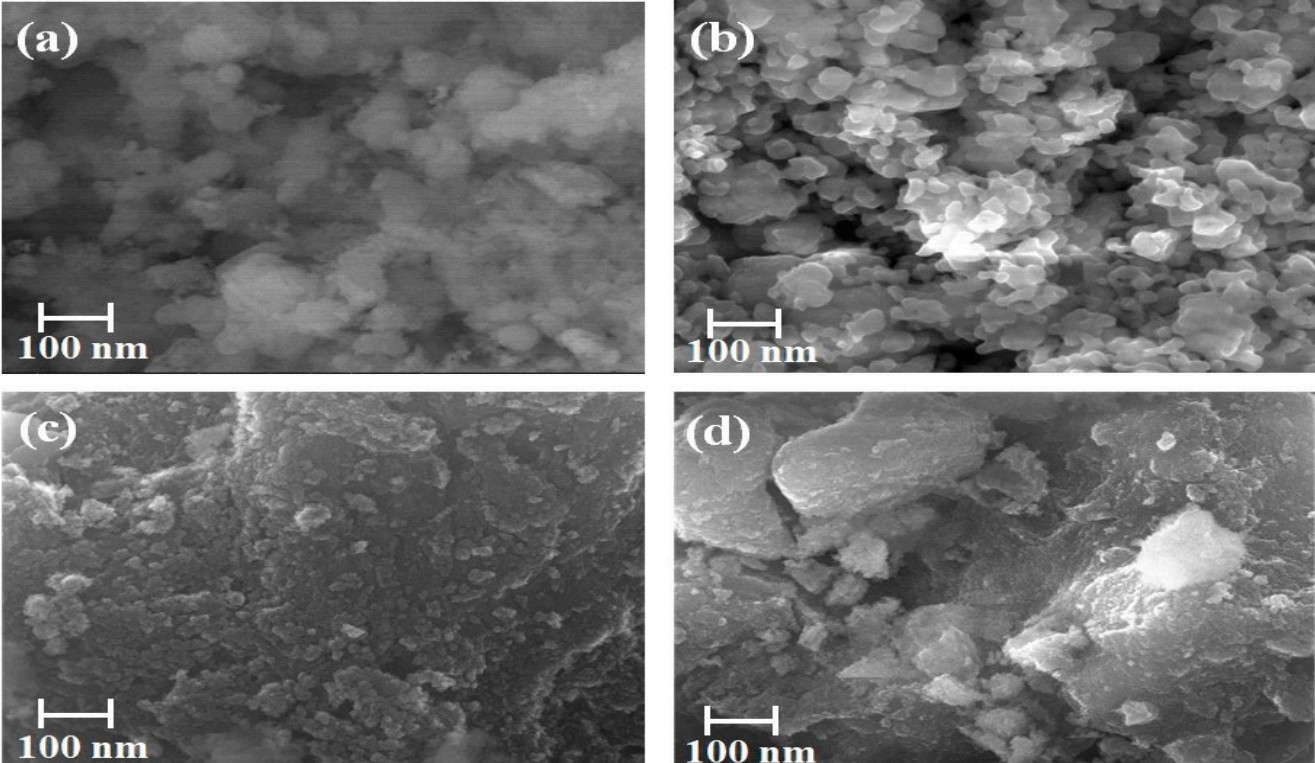

**Figure 7.** SEM images of (**a**) Cu/COK12, (**b**) Cu-$Nb_2O_5$, (**c**) Cu-Y-Zeolite, (**d**) Cu-ZSM5 catalyst.

3.1.5. Temperature Programmed Reduction (TPR)

Temperature-programmed reduction (TPR) is a technique that is widely used to examine metal oxide's surface chemistry under programmed temperature conditions. TPR yields quantitative and qualitative information about the reducing characteristics of the oxide's surface.

Figure 8 represents the consumption profiles of hydrogen in the $H_2$-TPR experiments of Cu-modified ZSM5, Y-Zeolite, $Nb_2O_5$, and COK12 catalysts. Except Cu-$Nb_2O_5$ (700–900 °C), all catalysts show reduction peak corresponding to CuO. Reduction of CuO include firstly CuO reduced to $Cu_2O$ then $Cu_2O$ to $Cu^0$ and strongly influence by the support, metal-support interactions, and copper dispersions [41]. Cu-COK12 and Cu-Y-Zeolite show single reduction peak indicate reduction of surface CuO. By contrast, the TPR profile of Cu-ZSM5 and Cu-$Nb_2O_5$ catalysts exhibits one additional shoulder peak suggesting reduction of surface copper oxides along with reduction of bulk copper oxide [42]. The peak shift to high temperature in Cu-ZSM5 and Cu-Y-Zeolite reflects the strong interaction between CuO and supports [43].

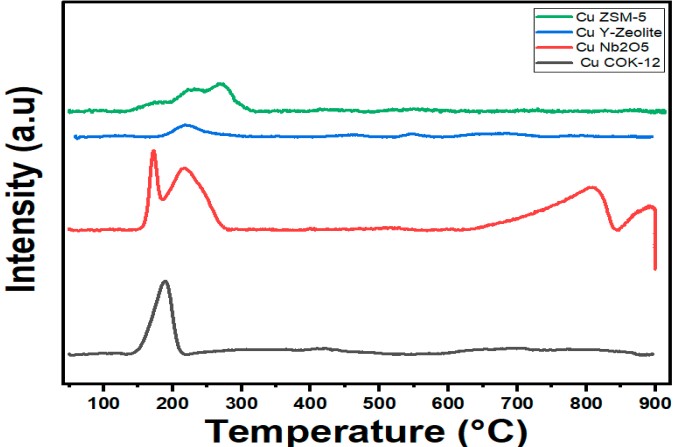

**Figure 8.** TPR profiles of Cu-modified ZSM5, $Nb_2O_5$, COK12, and Y-Zeolite catalysts.

3.1.6. Temperature Programmed Desorption (TPD)

TPD is a widely used technique to monitor the gas desorption from a solid surface while changing the investigated sample's temperature in a controlled manner. TPD can be used to evaluate active sites on the catalyst's surface in adsorption and catalytic reactions, understanding desorption, adsorption and surface reaction mechanisms, and acid/base properties of the solid sample. The sample is placed in a region where temperature can be changed. Usually, the sample temperature is varied linearly with time with the help of a temperature controller. The sample's surface is first exposed to a gas at a fixed temperature to obtain specific initial coverage, and sufficient time is allowed for the unabsorbed gas to flush out of the system. Small polar molecules like NH3, H2O, and $CO_2$ are usually used as adsorbate gases. The sample's heating provokes the evolution of adsorbed gases from the solid surface back into the gas phase. As the sample is heated, the adsorbed gas gets desorbed and is detected through detection devices. Various detectors used like Thermal Conductivity Detector (TCD), Flame Ionization Detector (FID), mass spectrometer, and conductometric titration, are used in TPD to detect and quantify the desorbed gas. Initially, the adsorption rate increases exponentially with increased temperature, attain a maximum and drops to zero as all the adsorbed gas gets desorbed. The data obtained from the TPD experiment is compiled into a plot between the detector signal intensity on the *y*-axis and time or temperature on the *x*-axis. The amount of adsorbate initially adsorbed onto the surface is proportional to the area under the TPD profile. The peak maxima position on the temperature scale is related to the heat of adsorption or activation energy

for desorption. The following two sections are dedicated for TPD analysis of catalysts being used in the study.

CO$_2$-Temperature Programmed Desorption (TPD)

CO$_2$ adsorption abilities or basicity of the COK12, Y-Zeolite, and ZSM5 supports and Cu modified COK12, Nb$_2$O$_5$, Y-Zeolite, and ZSM5 catalysts were studied by CO$_2$-TPD measurements, and the results are shown in Figure 9.

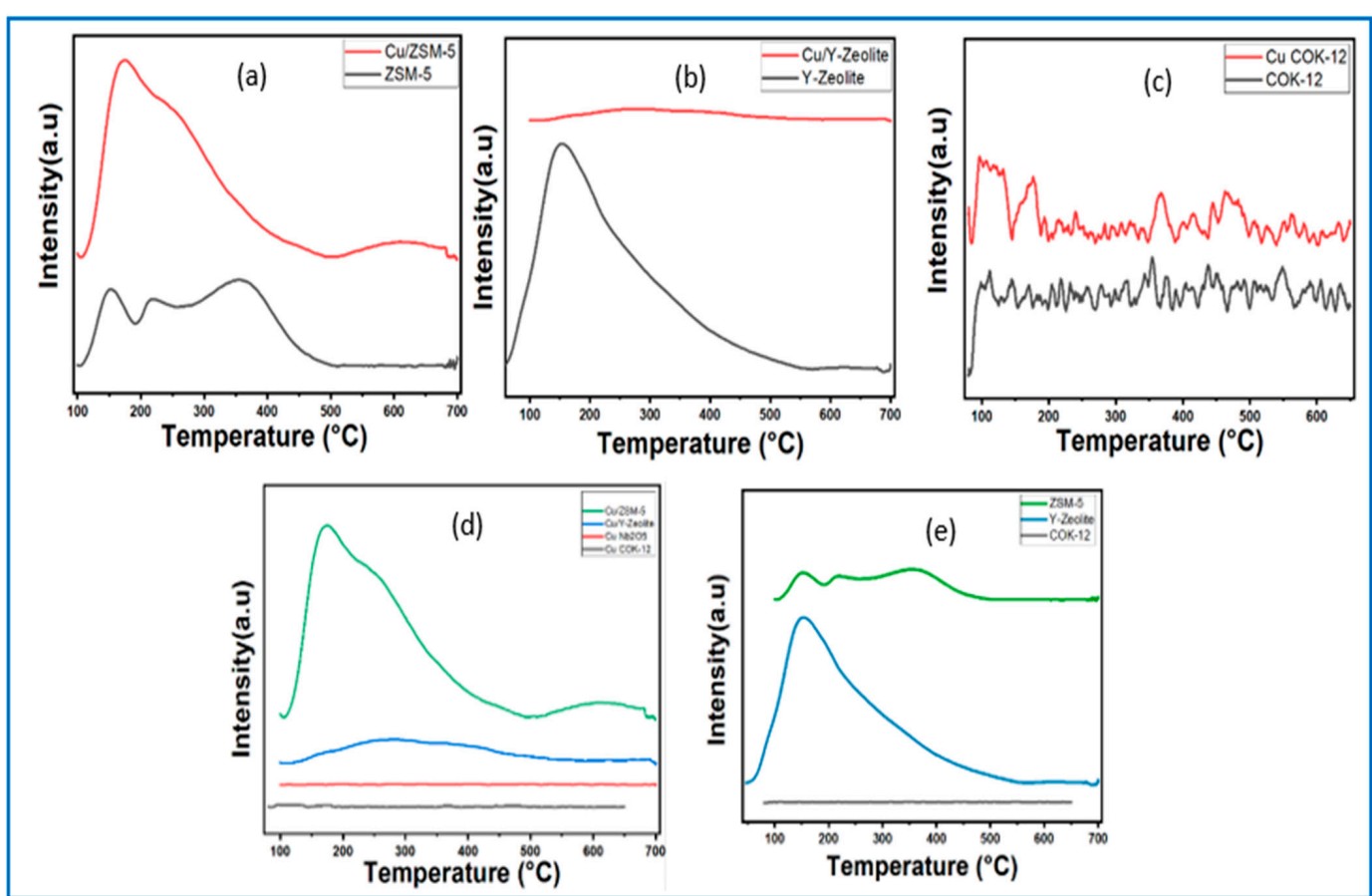

**Figure 9.** CO$_2$-TPD profiles of (**a**) Cu-ZSM5 and ZSM5, (**b**) Cu-Y-Zeolite and Y-Zeolite, (**c**) Cu/COK12 and COK12, (**d**) Cu modified COK12, Nb$_2$O$_5$, Y-Zeolite, and ZSM5 catalysts, and (**e**) ZSM5, Y-Zeolite, and COK12 support.

There was no desorption peak observed in COK12 and Cu-COK12, Cu-Nb$_2$O$_5$ catalysts. There was weak adsorption of CO$_2$ over Cu-Y-Zeolite catalyst in the temperature range 150–450 °C. The desorption curves of Cu decorated ZSM5 (Cu-ZSM5) catalyst showed two characteristic peaks corresponding to CO$_2$ adsorption on the intrinsic primary sites of Cu-ZSM5. The first peak was observed at a moderate temperature range (150–400 °C). In contrast, the second desorption peak was at a higher temperature up to about 450 °C that was ascribed to the desorption of weakly adsorbed CO$_2$

NH3-Temperature Programmed Desorption (TPD)

Figure 10 shows the NH3-TPD profiles of Cu backed on ZSM5, Y-Zeolite, Nb$_2$O$_5$, and COK12 catalysts and the corresponding support materials within the temperature extend of 50–700 °C. For Cu-Y-Zeolite, two desorption peaks were observed at 200 °C and 410 °C. The desorption peak at 200 °C is due to physiosorbed NH$_3$ or ammonium species, and the desorption peak at 410 °C is assigned to NH$_3$ held on the strong acid sites. After including the Cu, the NH3-TPD profile of H-ZSM5 has been changed and it is observed that when Cu

is included, there is a disappearance of one peak (shown within the graph). Two significant crests can be observed in the NH3-TPD profiles of Y-Zeolite, adjacent to each other, but due to Cu's addition within the location, three peaks were observed around 135, 400, 600 °C temperature within the chart. The Cu-$Nb_2O_5$ catalyst shows no ammonia desorption, but in $Nb_2O_5$, there is one peak around 257–262 °C.

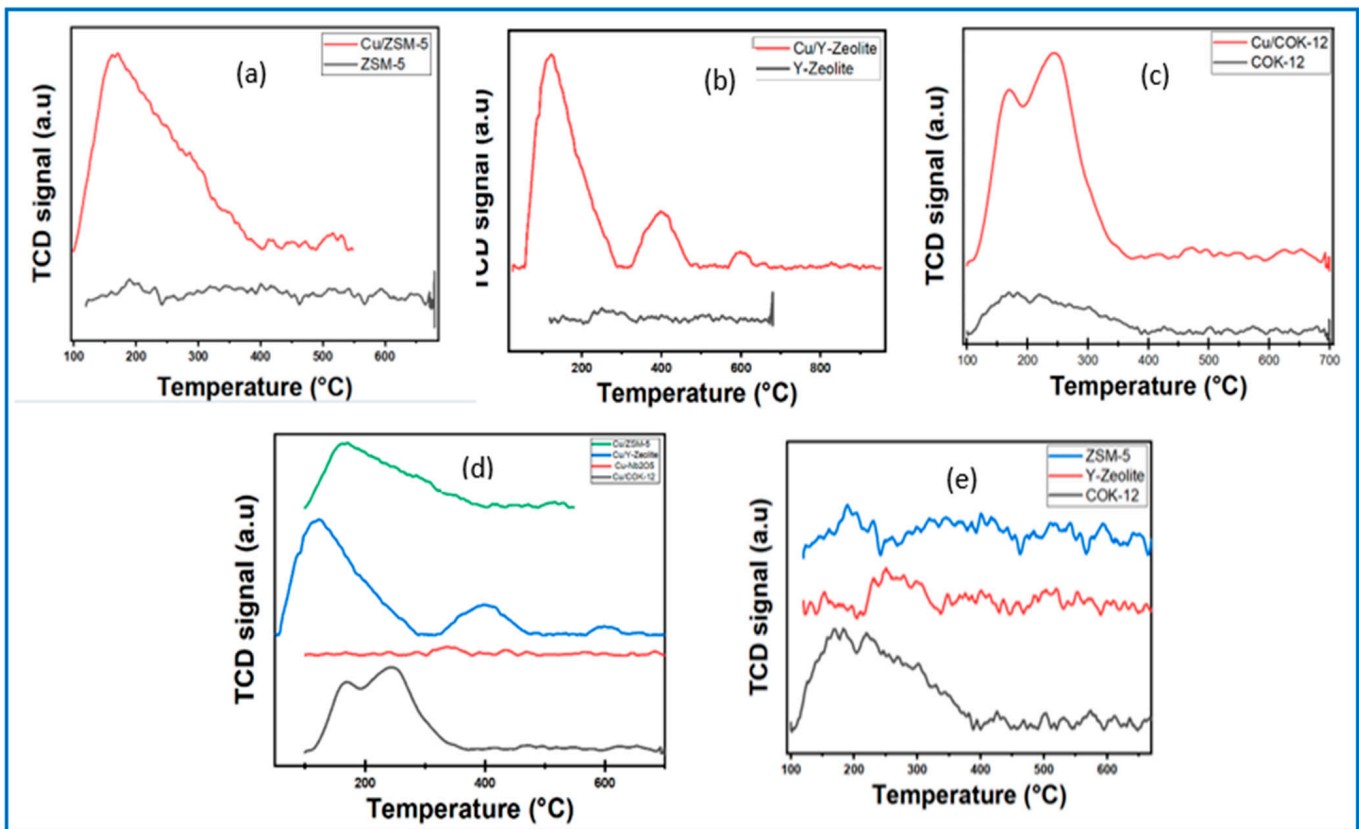

**Figure 10.** NH3-TPD profiles of (**a**) Cu-ZSM5 and ZSM5, (**b**) Cu-Y-Zeolite and Y-Zeolite (**c**) Cu/COK12 and COK12 (**d**) Cu mod-ified COK12, $Nb_2O_5$, Y-Zeolite, and ZSM5 catalysts, and (**e**) ZSM5, Y-Zeolite and COK12 support.

## 4. Application towards NO Conversion

This section aims to discuss the results of the NO conversion rate (%) of various supports alone as well support-active metal combinations. The multiple supports used are (i) COK12, (ii) $NB_2O_5$, (iii) Y-zeolite, and (iv) ZSM5. These supports were used with Cu, which is a non-noble active metal. With different combinations, a total of four catalysts were prepared: (i) Cu-COK12, (ii) Cu-$NB_2O_5$, (iii) Cu-Y-zeolite, and (iv) Cu-ZSM5.

Figure 11 represents the NO conversion rate for COK12, $Nb_2O_5$, Y-Zeolite, and ZSM5. The NO conversion rate was almost similar for COK12 and $Nb_2O_5$ for the entire temperature range. Negligible difference was observed i.e., 0.81, 6.72, 1.10, 5.69 and 1.28% at 200, 300, 400, 500 and 600 °C, respectively. Among Y-Zeolite and ZSM5, the later support showed promising results at low temperatures, i.e., 35.2, 29.2, 29.8% NO conversion rate at 200, 300, and 400 °C, respectively. The former catalyst showed promising results at high temperatures, i.e., 28.8 and 30.0% NO conversion rates at 500 and 600 °C temperatures. ZSM5 was found to be the best support based on the NO conversion rate.

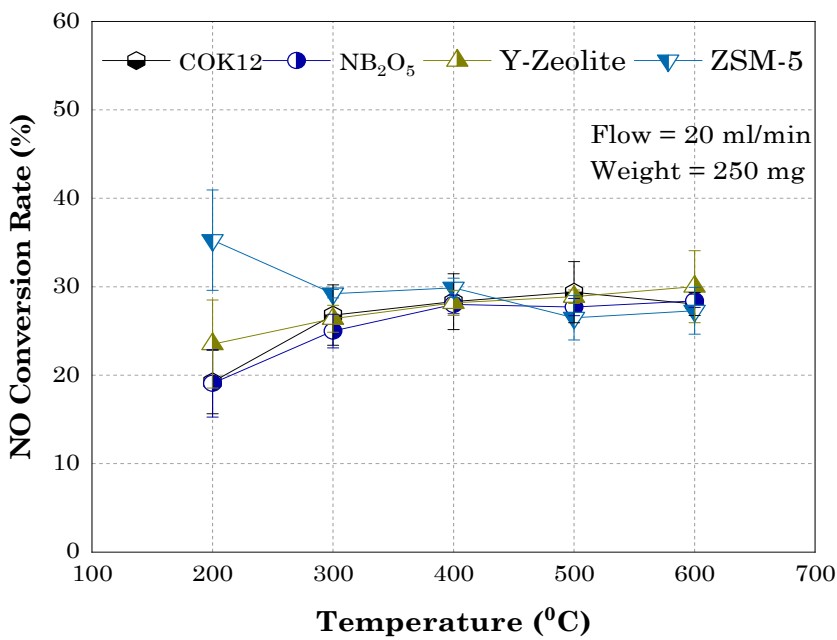

**Figure 11.** NO conversion rates vs temperature for various supports.

Figure 11 represents the NO conversation rate with temperature, for COK12 & Cu-COK12, $Nb_2O_5$ & Cu-$Nb_2O_5$, Y-Zeolite & Cu- Y-Zeolite and ZSM5 & Cu- ZSM5. Here, 250 mg catalyst was used, and the gas flow rate was maintained at 20 mL/min. Figure 12a represents the NO conversion rate for ZSM5 and Cu-ZSM5. Without Cu metal (ZSM5), NO conversion rate was ~35.2, 29.2, 29.8, 26.5, and 27.3% at temperatures of 200, 300, 400, 500, and 600 °C, respectively. With Cu doping (Cu-ZSM5) NO conversion rate jumped to ~40.6, ~34.2, ~35.5, 34.1 and 32.2% at 200, 300, 400, 500 and 600 °C temperature, respectively. Figure 12b represents the NO conversion rate for Y-Zeolite & Cu-Y-Zeolite. Without the use of active metal, Y-zeolite was able to convert NO with ~23.5, 26.3, 28.1, 28.8, and 30.0% conversion rate at 200, 300, 400, 500, and 600 °C, respectively. The combination of support and active metal (Cu- Y-Zeolite) enhanced NO conversion significantly compared to unloaded support, i.e., ~141.0, 117.2, 101.3, 106.5, 89.4% increase in conversion percentage at 200, 300, 400, 500, and 600 °C temperatures, respectively. Figure 12c represents the NO conversion rate for $Nb_2O_5$ and Cu- $Nb_2O_5$. NO conversion rate for $Nb_2O_5$ was ~19%, ~25%, ~28%, ~27.7%, ~28.4% at 200, 300, 400, 500 and 600 °C temperature, respectively. The influence of Cu presence was negligible in terms of NO conversion rate, especially at a higher temperature range. For Cu-$Nb_2O_5$ NO conversion rate was increased by ~64.2%, ~16.3%, ~3.9%, and ~9.3% at 200, 300, 400, and 500 °C, respectively. However, it was decreased by ~3.8% at 600 °C. Figure 12d represents the NO conversion rates for COK12 & Cu-COK12 with temperatures. It can be seen that without any active metal also, COK12 supports participates in the reaction and reduces NO. Pochamoni et al. [44] reported a similar type of results in nitrogen flow instead of NO gas. The NO conversion rate was ~19–29% in the entire temperature range, i.e., 200–600 °C. The NO conversion rate was increased to ~29% when support was tested at 300 °C. After that, the NO conversation rate was almost plateauing in the 300–600 °C temperature range. For COK12 support, the addition of Cu to COK12 enhanced the NO conversion rate significantly. It increased by ~207%, ~115%, ~107%, ~89% and ~107% at 200, 300, 400, 500, 600 °C respectively. The NO conversion rate was found higher than the conventional catalyst ($Yb_{0.50}$ $Tb_{0.50}$)2O$_3 \pm \delta$ (54%) [21].

Overall, it is evident that NO decomposition activity was significantly enhanced by introducing non-noble metal Cu. According to researchers around the globe, the Cu-ZSM5 is a well-established catalyst for NO decomposition [45–52], so further analysis would be focused on Cu-ZSM5.

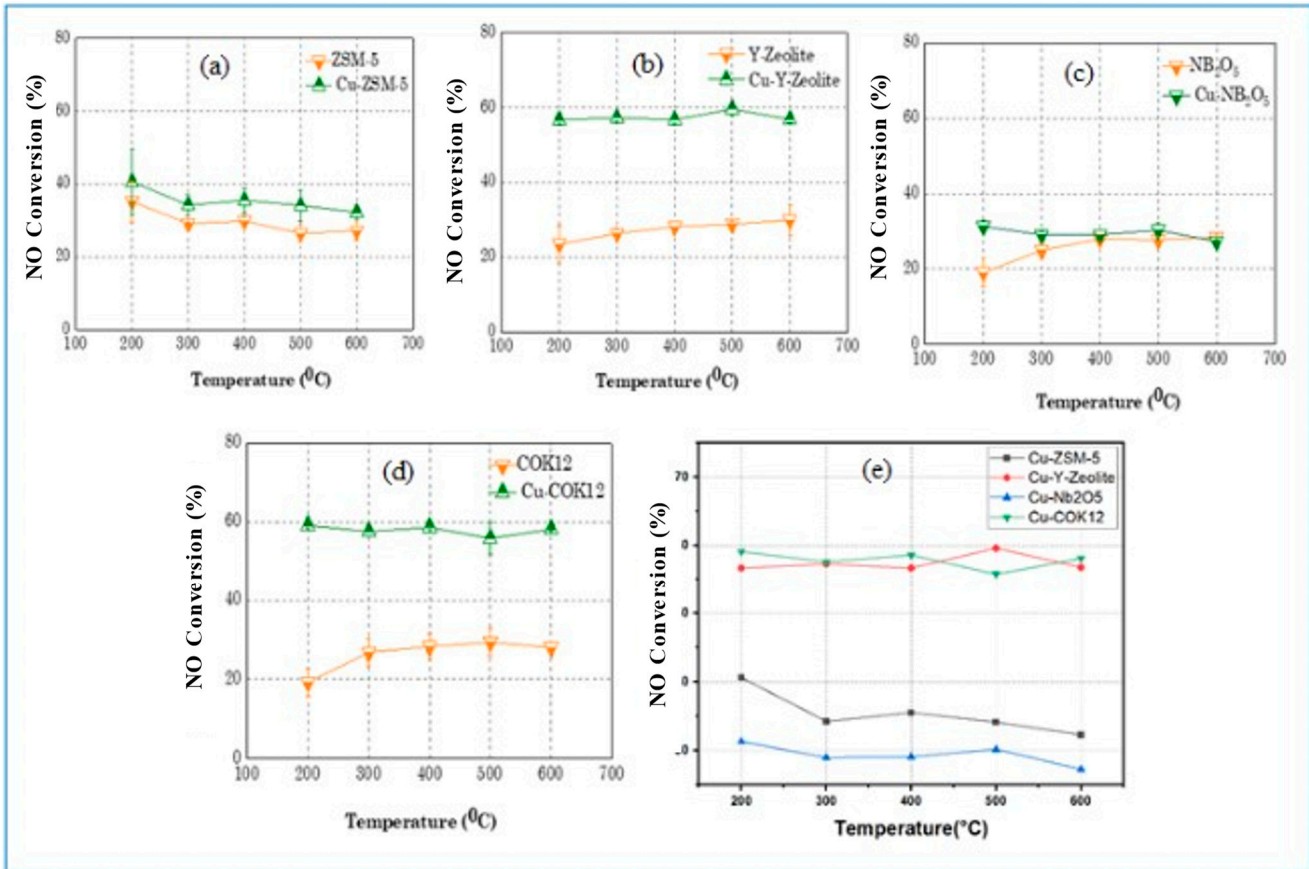

**Figure 12.** NO conversion rate (%) vs. Temperature (°C) for (**a**) ZSM5 & Cu-ZSM5, (**b**) Y-Zeolite & Cu-Y-Zeolite, (**c**) Nb$_2$O$_5$ and Cu- Nb$_2$O$_5$, (**d**) COK12 and Cu-COK12, and (**e**) Comparative plot for Cu over COK12, Nb$_2$O$_5$, Y-Zeolite, and ZSM5.

Figure 13 represents the NO conversion rate for Cu-ZSM5 at different temperatures (200–600 °C), the efforts were made to determine the influence of different exhaust flow rates, i.e., 15, 20, 50 mL/min on NO conversion, where 250 mg of catalyst was taken for the experiment. As discussed in the catalyst preparation Section 2.2, the achievement of 62.5% NO conversion is in coherence with the study performed by Curtin et al. [34], where they showed the NO conversion activity of 65%. The graphical representation of results in Figure 13 clearly shows that the 15 mL/min flow rate is the best as NO conversion rate was 55.9, 55.1, 51.4, 62.5, and 59.5% at temperatures of 200, 300, 400, 500, and 600 °C, respectively. For, 20 mL/min flow rate maximum and minimum NO conversion rate were 40.6 and 32.2% at 200 and 600 °C. At 50 mL/min flow rate, maximum and minimum conversion rates were 44.1 and 32.9% at 200 and 500 °C.

However, Figure 14 represents the NO conversion rate for Cu-ZSM5 at a constant flow rate of 20 mL/min and 250 mg catalyst on an hourly basis. The NO conversion activity of Cu-ZSM5 was observed for successive time intervals of 1 h at a constant temperature of 600 °C. The maximum and minimum NO conversion rate was observed for the 1st and 5th hour, i.e., 62.6 and 25.8%, respectively.

In the current research the space velocity is also considered as an important factor as NO conversion activity was also dependent on the space velocity, Figure 15 represents 62.5% as the highest NO conversion percentage by Cu-ZSM5 and a comparison with the literature, where at a relatively lower space velocity the NO conversion is higher with the same Cu-ZSM5 catalyst.

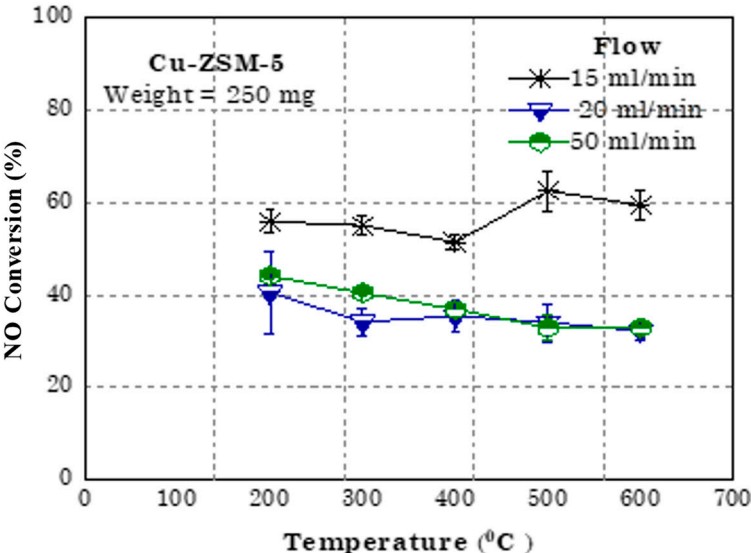

**Figure 13.** NO conversion rate (%) vs. temperature (°C) for Cu-ZSM5.

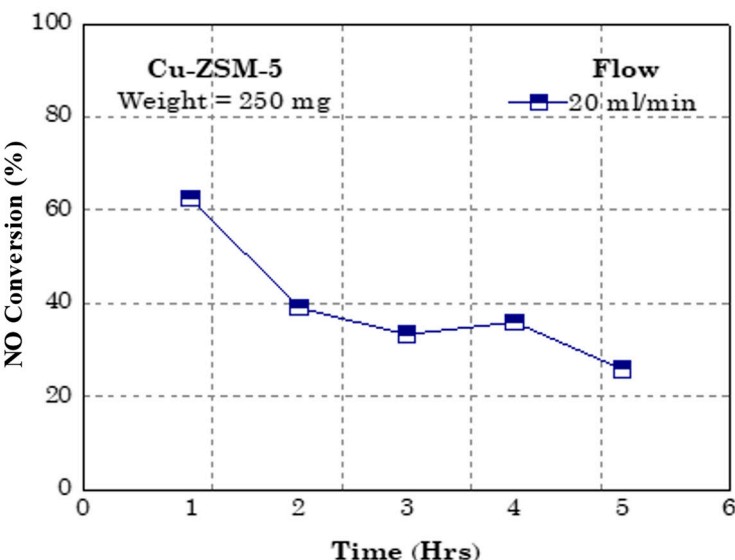

**Figure 14.** NO conversion rate (%) vs. Time (Hrs) for 20 mL/min flow rate and 600°C temperature.

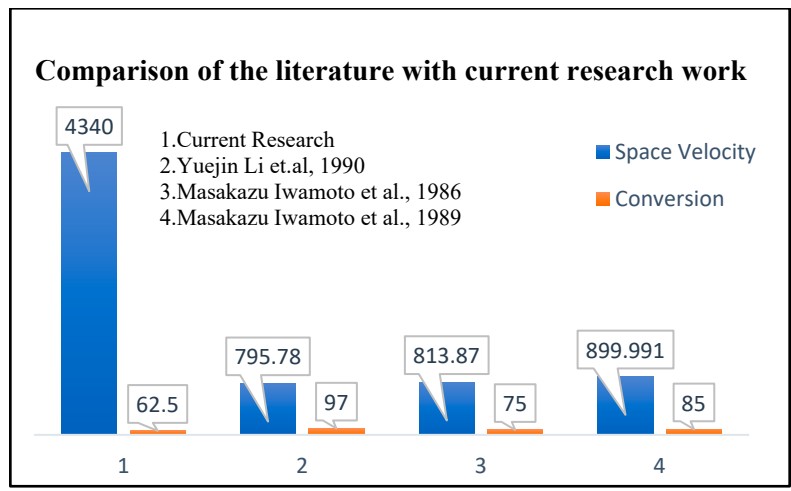

**Figure 15.** Comparison of space velocity and NO conversion [53–55].

## 5. Conclusions

This research has added to the study of the direct decomposition of NO with the help of non-noble metal Cu doped on COK12, $Nb_2O_5$, Y-Zeolite, and ZSM5 support, with an aim to produce a low-cost solution for NO abatement. The catalysts Cu-COK12, Cu-$Nb_2O_5$, Cu-Y-Zeolite, and Cu-ZSM5 were produced by the wet impregnation method. The BET results in combination with the SEM, TEM, and $H_2$-TPR analysis indicated that CuO was highly dispersed on the external surface and deposited inside the pores of the Cu-modified ZSM5 Y-Zeolite, $Nb_2O_5$, and COK12 catalysts, which is in good agreement with previous reports. Additionally, the characterization techniques such as XRD, SEM, TEM, TPR, $CO_2$ TPD, and $NH_3$ TPD well established the prepared catalysts' suitability and potential to be used for NO's direct decomposition. Furthermore, an experimental investigation has been performed to show the reactivity of COK12, $Nb_2O_5$, Y-Zeolite, and ZSM5 towards NO decomposition.

Further reactivity analysis was performed on Cu-COK12, Cu-$Nb_2O_5$, Cu-Y-Zeolite, and Cu-ZSM5. The comparative analysis concluded that adding Cu with the chosen supports has increased the NO decomposition capability. The idea for 3% Cu addition came from Curtin et al. [34] who did experimental investigations for direct decomposition of NO using Cu-ZSM5 catalyst with varying weight percentages of Cu in the catalyst, they found that for weight percentages of Cu from 2.5% to 3% the NO conversion was 65% quoted as maximum. Moreover, a separate experimental analysis on Cu-ZSM5 catalyst showed that for a fixed catalyst weight and in a definite temperature range, varying the flow rates of NO could impact the NO decomposition activity. The Cu-ZSM5 is a well-established catalyst for NO conversion so further experimental trials were performed. The important interpretation of this experimental analysis can be stated as; for a lower flow rate of NO, the NO decomposition activity of the catalyst is higher, and for higher flow rates, the reactivity of the catalyst is hampered. Additionally, the results of experiments in this research work serves a clear idea of trends of NO conversion for given catalysts at the given conditions. In the future in situ infrared characterization experiments are planned on all the catalysts, as to conclude on the reaction process and mechanisms the active medium species are really significant. In the future, it is intended to increase the NO conversion percentages; our focus is to check the relation between the reactivity of Cu-COK12, Cu-$Nb_2O_5$, Cu-Y-Zeolite catalysts, and flow rates of NO gas, and further investigations would also consider the variation in catalyst weight, and hence to represent the catalytic reactivity towards NO decomposition in terms of space velocity.

**Author Contributions:** M.K.S.: conceptualization, methodology, visualization, draft preparation; B.V.S.C.: investigation, draft preparation, writing and editing; S.V.: conceptualization, reviewing and editing the draft; A.D.: validation, supervision. All authors have read and agreed to the published version of the manuscript.

**Funding:** The project was funded by CSIR, project number OLP 101919.

**Institutional Review Board Statement:** There is no human study involved so ethical approval was not required in the study.

**Informed Consent Statement:** Not applicable.

**Acknowledgments:** The preparation of catalysts, characterization of catalysts, and experimental investigations was performed at CSIR-Indian Institute of Petroleum, Dehradun, UK, India. The work was supervised by Atul Dhar, School of Engineering, Indian Institute of Technology Mandi, H.P., India. The research was performed with the collaboration of CSIR-Indian Institute of Petroleum, Dehradun, UK, India; Indian Institute of Technology Mandi, H.P., India; Centre of Earth Observation Science, University of Brighton, UK; and City, University of London, UK.

**Conflicts of Interest:** The authors declare no conflict of interest.

**Sample Availability:** Samples of the compounds are available from the authors.

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
