# Peer review of "Catalytic Direct Decomposition of NOx Using Non-Noble Metal Catalysts"

_solids, doi:10.3390/solids3040041_

Round 1

Reviewer 1 Report

Attached please find the comments.

Author Response

Reviewer 1

Comments on Solids-1873697:

In this paper, an experimental investigation has been performed to show the reactivity of various supports alone as well support-active metal combinations. Overall, the article is well organized, and its presentation is good. However, some minor issues still need to be improved.

  1. BJH can be analyzed in detail from two aspects of pore volume and pore size to highlight the advantage of Cu-ZSM-5. Please make some additions.

Answer: Thank you very much for your comment to improve our manuscript, we have drawn the BJH curves for all the catalyst and added them in the manuscript after your suggestions. The pore size for ZSM5 and Cu-ZSM is mentioned. (Please refer to line number 268)

The updated version of the section is as follows:

Catalyst

Surface area, m2/g

Pore volume, cm3/g

Pore Size, nm

COK-12

270

0.79

11.80

3%Cu/COK-12

323

0.72

8.90

Nb2O5

15.4

0.06

16.00

3%Cu/Nb2O5

6.5

0.08

53.6

Y-Zeolite

325

0.47

7.45

3%Cu/Y-Zeolite

230

0.50

8.70

ZSM-5

293

0.20

5.90

3%Cu/ZSM-5

228

0.37

6.50

It is observed that the surface area is increased by 20% in the case of COK-12 when modified with Cu. The pore size & pore volume was also decreased by 24% which suggests that the Cu species couldn’t penetrate to the pores of COK-12 but adsorbed over the COK-12 surface. Whereas, in the case of Cu-modified Nb2O5, Y-Zeolite & ZSM-5 the pore volume & pore size is increased, and the surface area decreased in all the Cu-modified catalysts. The pore volume & pore size is increased by 85% & 32% respectively in Cu/ZSM-5 compared to ZSM-5 which indicates a good level of penetration to the pores of ZSM-5 by Cu species is occurred by changing the channels structure of ZSM-5. In the case of Cu-modified Y-Zeolite & Nb2O5, the pore size increased by 50% & 235% respectively, but the pore volume is increased by 6 % & 33 % only. It indicated that the pores of Y-Zeolite & Nb2O5 are almost covered when modifying with Cu, but in the case of Cu/ZSM5, Cu is not covering the pores of ZSM5 & thus keeping the pores available for the reaction to happen. It may be attributed to the fact that after entering the pores, the Cu might break the Si-Al network in the channels of ZSM-5 & formed its bond within ZSM-5 network, making the larger pore volume & pore size. Due to this, the reactant will have to react first with the Si-Al layer & then with the Cu particle. It will avoid the increment of Cu particle size during reaction & thus agglomeration & sintering. Although the surface area of ZSM5 is decreased with the addition of Cu, but a good conversion & a larger pore volume is worth considering when compared with the decrease in surface area from 293 m2/g to 228 m2/g., the addition of Cu with ZSM5 resulted in a decrease in surface area; this can also be due to the fact that CuO particles (generally the copper species) may have partially covered the pores of the ZSM5's external surface or the channels. The same trend was observed in the case of Cu-modified Y-Zeolite, Nb2O5, and COK-12 catalysts.

  1. The TPD is only briefly described, but further analysis is expected.

Answer: Thank you very much for pointing this, authors has now added extended research on TPD, which is given as follows:

TPD is a widely used technique to monitor the gas desorption from a solid surface while changing the investigated sample’s temperature in a controlled manner. TPD can be used to evaluate active sites on the catalyst’s surface in adsorption and catalytic reactions, understanding desorption, adsorption and surface reaction mechanisms, and acid/base properties of the solid sample. The sample is placed in a region where temperature can be changed. Usually, the sample temperature is varied linearly with time with the help of a temperature controller. The sample's surface is first exposed to a gas at a fixed temperature to obtain specific initial coverage, and sufficient time is allowed for the unabsorbed gas to flush out of the system. Small polar molecules like NH3, H2O, and CO2 are usually used as adsorbate gases. The sample’s heating provokes the evolution of adsorbed gases from the solid surface back into the gas phase. As the sample is heated, the adsorbed gas gets desorbed and is detected through detection devices. Various detectors used like Thermal Conductivity Detector (TCD), Flame Ionization Detector (FID), mass spectrometer, and conductometric titration, are used in TPD to detect and quantify the desorbed gas. Initially, the adsorption rate increases exponentially with increased temperature, attain a maximum and drops to zero as all the adsorbed gas gets desorbed. The data obtained from the TPD experiment is compiled into a plot between the detector signal intensity on the y-axis and time or temperature on the x-axis. The amount of adsorbate initially adsorbed onto the surface is proportional to the area under the TPD profile. The peak maxima position on the temperature scale is related to the heat of adsorption or activation energy for desorption. Temperature programmed desorption (TPD) studies were carried out using an Autochem HP II-2950 chemisorption analyzer (Micromeritics, USA). NH3-TPD was performed on Micromeritics, Autochem HP II-2950, equipped with a thermal conductivity detector (TCD). Before analysis, the sample was pre-treated at 300 ˚C for 1 h in He, and then it was exposed to 30 ml.min-1 flow of 10% NH3-He for 30 min. After the adsorption, it was exposed to He for 30 min to remove extra NH3-He present over the surface; then, the temperature was raised to 700 ˚C for TPD measurements. The following two sections are dedicated for TPD analysis of catalysts being used in the study.

  1. XRD are marked with detailed diagrams to facilitate readers to better understand your expression, and it will be more professional.

Answer: Thank you very much for your comment.

  1. Introduction mentions deficiencies in previous researchplease supplement the performance experiment of various catalysts in the presence of water to make conclusions more convincing.

Answer: Thank you very much for your words to improve our manuscript. The performance experiment of catalysts is done in the quartz glass reactor bed, which is placed in a furnace with temperature varying from 200°C to 600°C, maintained for testing the catalyst performance in conditions similar to the exhaust tail (catalytic converter) of engines. In the line number 179-181 we quoted that the experiments on Cu-ZSM5 in presence of water has already been performed in literature, the Cu/ZSM-5 catalyst has poor activity in the presence of water, which sometimes changes the chemical reaction such that it affects the redistribution of the Cu2+ ions over a zeolite catalyst.

  1. There are a few typos and grammar errors in this paper. It will be good to do a thorough proof-reading of the manuscript again.

Answer:  Thank you very much for pointing out the errors, we have thoroughly proofread the paper and removed the typos and grammar errors to the best of our knowledge.

  1. Some new literatures might be helping the authors to further deepen the understanding of reaction mechanism as well as newest developing in this field (Bioresource Technology, 2021, 332: 125086 Study on the Hg0 removal characteristics and synergistic mechanism of iron-based modified biochar doped with multiple metals).

Answer: Thank you very much for your suggestion, we have added new literature for reaction mechanism, and the suggested article was quite well and worth reading, so we have cited that paper and some similar literature. (Please refer to line number 221)

Jia et al. also used a fixed bed reactor setup for the removal of Hg0 from the modified biochar, later did a temperature programmed desorption analysis [35]. The quartz glass reactor idea is followed from the studies of S.B. Jorgensen [36] and Stegehake et al.,[37] where the modelling and validation of fixed bed reactors, and their applicability for reaction set-up is praised.

We believe that this experimental research analysis forms a strong basis for future research on catalytic Direct Decomposition of NOx using Non-Noble Metal Catalysts such as Cu-COK-12, Cu-Nb2O5, Cu-Y-Zeolite, and Cu-ZSM5. We will be highly obliged to get your consideration and support to add our research in this field.

Reviewer 2 Report

The catalytic decomposition of NO is an effective method for the abatement of NO. This paper investigated the catalytic decomposition of NO on copper ion-exchanged with different bases such as COK-12, Nb2O5, Y-Zeolite, and ZSM5. The comparative experimental analysis has shown that adding Cu with COK-12, Nb2O5, Y-Zeolite, and ZSM5 supports gave a precedented rise in NO decomposition compared to stand-alone supports. The analysis of the characterization for the different catalysts were carried out. However, there are also some important problems in the manuscript. The reviewer think it can be published in the journal of SOLIDS after major revisions.  The detailed suggestions are listed as follows:

1. In the activity analysis part, the author descripted that “the activity of the catalyst can be only improved significantly by adding Cu”. But the appropriate amount of Cu should be optimized through experiments and investigated the activity.

2.What is the main reason for improved activity by adding Cu. The intrinsic reason was not presented in the paper. If you could further elaborate it, the conclusion would be more convincing.

3.Since the selectivity still remains as the biggest issue of NO decomposition reaction on catalysts. I wonder if the influences of different supports for the N2 selectivity.

4. The space velocity is very important for NO decomposition reaction. The author should present the space velocity. how did the author choose the rate of the flow? I think it was very slow.

5.The active medium species are important for understanding the reaction process and concluding the reaction mechanism. Therefore, it is suggested that In situ infrared characterization experiments should be carried out on the different catalysts.

6.The author should study on structure-activity relationship on the different catalysts.

7.The author should add the computational formulas of NO conversion and N2 selectivity in the beginning.

Author Response

Reviewer 2

The catalytic decomposition of NO is an effective method for the abatement of NO. This paper investigated the catalytic decomposition of NO on copper ion-exchanged with different bases such as COK-12, Nb2O5, Y-Zeolite, and ZSM5. The comparative experimental analysis has shown that adding Cu with COK-12, Nb2O5, Y-Zeolite, and ZSM5 supports gave a precedented rise in NO decomposition compared to stand-alone supports. The analysis of the characterization for the different catalysts were carried out. However, there are also some important problems in the manuscript. The reviewer think it can be published in the journal of SOLIDS after major revisions.  The detailed suggestions are listed as follows:

  1. In the activity analysis part, the author descripted that “the activity of the catalyst can be only improved significantly by adding Cu”. But the appropriate amount of Cu should be optimized through experiments and investigated the activity.

Answer: Thank you very much for your view to improve our manuscript, the reason for adding an appropriate 3% of Cu being used on the supports COK-12, Nb2O5, Y-Zeolite, and ZSM5 is now quoted in the manuscript. We have gone through a literature given by Curtin et al., who did experimental investigations for direct decomposition of NO using Cu-ZSM5 catalyst with varying weight percentages of Cu in the catalyst, they found that for weight percentages of Cu from 2.5% to 3% the NO conversion was 65% quoted as maximum. A new reference has now been added to support the statement in section 2.2 Catalyst preparation.

  1. What is the main reason for improved activity by adding Cu. The intrinsic reason was not presented in the paper. If you could further elaborate it, the conclusion would be more convincing.

Answer: Thank you very much for your comment, the reason for adding Cu has been taken from literature, and reason for why exactly 3% of Cu was added is also present in the manuscript now:

The rationale for the addition of 3% Cu is based on the literature i.e. COK-12, Nb2O5, Y-Zeolite, and ZSM5 is being used by Curtin et al., [34] they did experimental investigations for direct decomposition of NO using Cu-ZSM5 catalyst with varying weight percentages of Cu in the catalyst, they found that for weight percentages of Cu from 2.5% to 3% the NO conversion was optimum and as maximum (65%).

  1. Since the selectivity still remains as the biggest issue of NO decomposition reaction on catalysts.I wonder if the influences of different supports for the N2

Answer: Thank you very much for your comment. Yes, the rate of support on the selectivity will significantly change due to strong metal support interactions.

  1. The space velocity is very important for NO decomposition reaction. The author should present the space velocity. how did the author choose the rate of the flow? I think it was very slow.

Answer: Thanks for your comment on the space velocity. We have now added the space velocity used in our research and compared it with some literature as shown in the figure 15. The space velocity used in our research is high as compared to the literature, they quoted that at lower space velocities, the NO conversion by Cu-ZSM5 is higher. We have planned that in the future work we are going to work on our experimental setup to reduce the space velocity in order to raise the NO conversion activity of catalysts.

  1. The active medium species are important for understanding the reaction process and concluding the reaction mechanism. Therefore, it is suggested that in situ infrared characterization experiments should be carried out on the different catalysts.

Answer: Thank you very much for your suggestion. Due to analytic limitation and the identification of reaction intermediates for the direct decomposition of NOx is well established in the open literature hence not considered in the study.

  1. The author should study on structure-activity relationship on the different catalysts.

Answer: Thank you very much for your suggestion. The structure activity relationship will be studied in our next project.

  1. The author should add the computational formulas of NO conversion and N2selectivity in the beginning.                            

Answer: Thank you very much for your suggestion. The structure activity relationship will be studied in our next project.

Reviewer 3 Report

The authors have presented the direct NOx decomposition over non-noble metal-based catalysts. Despite interesting results, the article has many technical flaws; hence, I do not recommend the publication of this article, in its current form, in Solids due to following reasons:

1.     The writing needs a complete revision both from the English language perspective as well as typos, subscript/superscript, units of quantities etc.

2.     The Abstract needs to be revised carefully. Abbreviations need to be defined at their first appearance. The redundant and unnecessary information needs to be removed.

3.     Lines 42-44, reference should be added.

4.     Line 45, please revise the sentence starting from “Since the transport sector….” Furthermore types of NOx need to be referenced.

5.     The authors need to revise introduction, cite references, and follow reference format of Solids. In addition the redundant information leads to loss of reader’s interest.

6.     The experimental details of TPD/TPR is missing.

7.     Lines 261-265, authors claimed that a decrease in specific surface area after copper loading was observed for all the catalysts but 3%Cu/COK-12 showed otherwise. Why?

8.     What is the purpose of adding Figure 2e and 2f? Moreover, the authors claimed in line270-272 that CuO is not detected but contradict themselves in lines 275-283?

9.     No data interpretation and discussion is provided for TPR results.

10.  Why authors added Figure 9d and 9e? Why there is no peak after loading Cu over COK-12 and Nb2O5? No data interpretation and data analysis?

11.  Why there is noise and different temperature ranges for NH3-TPD?

12.  Figures 11-14, y-axis should be conversion instead of conversion rate. The x-axis units need to be same.

13.  Why did author choose Cu-ZSM5 for long term stability test?

14.  Conclusions are required to be revised.

15.  Overall authors need to significantly revise and improve the content and presentation of the article in order to get it published in Solids.

Author Response

Reviewer 3

The authors have presented the direct NOx decomposition over non-noble metal-based catalysts. Despite interesting results, the article has many technical flaws; hence, I do not recommend the publication of this article, in its current form, in Solids due to following reasons:

  1. The writing needs a complete revision both from the English language perspective as well as typos, subscript/superscript, units of quantities etc.

Answer: Thank you very much for your comment on the English corrections, typos, subscript/superscripts, we have thoroughly checked the manuscript to the best of our knowledge.

  1. The Abstract needs to be revised carefully. Abbreviations need to be defined at their first appearance. The redundant and unnecessary information needs to be removed.

Answer: Thank you very much for your comment to improve the abstract, we have revised the manuscript and unnecessary information is plucked out from the draft. Abbreviations are carefully checked again. We have removed the following lines:

Internal combustion engines can be divided into two types of engines for vehicular applications: (i) Spark Ignition (SI) engines and (ii) Compression Ignition (CI) engines. The engine cylinders where an electric spark initiates the ignition of mixtures (air and fuel) are SI engines (also known as petrol engines).

 In these engines, fuel can be injected in port using a carburetor and a fuel injector. Another engine type is compression ignition engines (also known as diesel engines as Rudolph Diesel has invented this engine).

Fuel is also injected directly into the combustion chamber, at the end of the compression stroke, and combustion is initiated by hot combustion air.

Diesel engines are more efficient than petrol engines. However, diesel engines produce more NOx than petrol engines as they operate at relatively high temperatures and pressures, favoring NOx formation.

  1. Lines 42-44, reference should be added.

Answer: The following references are added to support the statements in line 42-44:

For the first statement (NOx is the togetherness of two brutal pollutants, i.e., NO and NO2):

  1. Ravina, M., Caramitti, G., Panepinto, D. and Zanetti, M., Air quality and photochemical reactions: analysis of NOx and NO2 concentrations in the urban area of Turin, Italy. Air Quality, Atmosphere & Health, 2022. 15(3), pp.541-558.
  2. Wang, J., Alli, A.S., Clark, S., Hughes, A., Ezzati, M., Beddows, A., Vallarino, J., Nimo, J., Bedford-Moses, J., Baah, S. and Owusu, G., Nitrogen oxides (NO and NO2) pollution in the Accra metropolis: Spatiotemporal patterns and the role of meteorology. Science of the Total Environment, 803, p.149931.

For the second statement(NOx is responsible for red line problems such as ozone layer depletion, acid rain, photochemical smog, tropospheric ozone, and global warming).

  1. Pearson, J.K. and Derwent, R., Air Pollution and Climate Change: The Basics. Routledge 2022.
  2. Mohite, V.T., 2022. Pollution and Pollution Control. In Emerging Trends in Environmental Biotechnology, 2022 (pp. 11-21). CRC Press.
  3. Rathnasekara, P.K. and Gunasekera, M.Y., Chemical process route selection based upon potential environmental risk of chemical releases. Results in Engineering, p.100589.

For the statement (The two primary sources of NOx are industries and the transport sector)

  1. Bharti, S., Chauhan, B.V., Garg, A., Vedratnam, A. and Shukla, M.K., Potential of E-Fuels for Decarbonization of Transport Sector. In Greener and Scalable E-fuels for Decarbonization of Transport, 2022, (pp. 9-32). Springer, Singapore.

For the statement (Since the transport sector is widespread and uses small bore and large bore engine driven vehicles, it can be considered as significant source of NOx emissions):

  1. Chauhan, B.V., Sayyed, I., Vedratnam, A., Garg, A., Bharti, S. and Shukla, M., 2022. State of the Art in Low-Temperature Combustion Technologies: HCCI, PCCI, and RCCI. Advanced Combustion for Sustainable Transport, pp.95-139.
  2. Garg, A., Chauhan, B.V., Vedratnam, A., Jain, S. and Bharti, S., 2022. Potential and Challenges of Using Biodiesel in a Compression Ignition Engine. Potential and Challenges of Low Carbon Fuels for Sustainable Transport, pp.289-317.

  1. Line 45, please revise the sentence starting from “Since the transport sector….” Furthermore types of NOx need to be referenced.

Answer: Thank you very much for your comment, yes there was a mistake in this line, it is revised and written correctly as

“ Since the transport sector is widespread and uses small bore and large bore engine driven vehicles, it can be considered as significant source of NOx emissions.”

The types of NOx are referenced by:

Cellek, M.S., The decreasing effect of ammonia enrichment on the combustion emission of hydrogen, methane, and propane fuels. International Journal of Hydrogen Energy 2022.

  1. The authors need to revise introduction, cite references, and follow reference format of Solids. In addition, the redundant information leads to loss of reader’s interest.

Answer: Dear reviewer, thank you for suggestions to improve our manuscript, the introduction is revised, and references are cited in the format of Solids. All the unnecessary lines are removed.

  1. The experimental details of TPD/TPR are missing.

Answer: The experimental details of TPD and TPR are added as follows, a figure of our TPD instrument is also added.

For TPR:

Temperature-programmed reduction (TPR) is a technique that is widely used to examine metal oxide’s surface chemistry under programmed temperature conditions. TPR yields quantitative and qualitative information about the reducing characteristics of the oxide’s surface. The Micromeritics Auto Chem II apparatus was used to carry out the temperature-programmed reduction experiments. The sample was placed in the TPR cell and flushed with argon for 30 min at 150oC. The sample was then subsequently cooled down to room temperature. Finally, the furnace temperature was increased at a ramp-up rate of 10°C/min in a 15 ml/min flow rate with the H2/Ar mixture (10:90 ratios). A thermal conductivity detector (TCD) was used to monitor the signals corresponding to H2 consumption.

For TPD:

TPD is a widely used technique to monitor the gas desorption from a solid surface while changing the investigated sample’s temperature in a controlled manner. TPD can be used to evaluate active sites on the catalyst’s surface in adsorption and catalytic reactions, understanding desorption, adsorption and surface reaction mechanisms, and acid/base properties of the solid sample. The sample is placed in a region where temperature can be changed. Usually, the sample temperature is varied linearly with time with the help of a temperature controller. The sample's surface is first exposed to a gas at a fixed temperature to obtain specific initial coverage, and sufficient time is allowed for the unabsorbed gas to flush out of the system. Small polar molecules like NH3, H2O, and CO2 are usually used as adsorbate gases. The sample’s heating provokes the evolution of adsorbed gases from the solid surface back into the gas phase. As the sample is heated, the adsorbed gas gets desorbed and is detected through detection devices. Various detectors used like Thermal Conductivity Detector (TCD), Flame Ionization Detector (FID), mass spectrometer, and conductometric titration, are used in TPD to detect and quantify the desorbed gas. Initially, the adsorption rate increases exponentially with increased temperature, attain a maximum and drops to zero as all the adsorbed gas gets desorbed. The data obtained from the TPD experiment is compiled into a plot between the detector signal intensity on the y-axis and time or temperature on the x-axis. The amount of adsorbate initially adsorbed onto the surface is proportional to the area under the TPD profile. The peak maxima position on the temperature scale is related to the heat of adsorption or activation energy for desorption. Temperature programmed desorption (TPD) studies were carried out using an Autochem HP II-2950 chemisorption analyzer (Micromeritics, USA). NH3-TPD was performed on Micromeritics, Autochem HP II-2950, equipped with a thermal conductivity detector (TCD). Before analysis, the sample was pre-treated at 300 ˚C for 1 h in He, and then it was exposed to 30 ml.min-1 flow of 10% NH3-He for 30 min. After the adsorption, it was exposed to He for 30 min to remove extra NH3-He present over the surface; then, the temperature was raised to 700 ˚C for TPD measurements. The following two sections are dedicated for TPD analysis of catalysts being used in the study.

.

  1. Lines 261-265, authors claimed that a decrease in specific surface area after copper loading was observed for all the catalysts but 3%Cu/COK-12 showed otherwise. Why?

Answer: Thank you for your comment, the section is now re-written, and the reason has also been quoted now. Many thanks for pointing this.

  1. What is the purpose of adding Figure 2e and 2f? Moreover, the authors claimed in line270-272 that CuO is not detected but contradict themselves in lines 275-283?

Answer: Figures 2e and 2F have been deleted as suggested. The text has been modified for clarity. The new text is “All typical peaks were compared with the XRD range of reported ZSM-5, Y-Zeolite, Nb2O5, and COK-12 catalyst. The Cu-ZSM5 catalysts’ diffraction peaks were seen at 8.9°, 22.9°, 23.8°, and 24.3°, which are corresponding to Miller indices of ZSM-5 support. Similarly in case of Cu/Y-Zeolite, typical peaks were observed at around 43° (112) and 68° (311) of Y-Zeolite. Further Cu/Nb2O5 catalyst shows Nb2O5 peaks designated as (111), (020), (111), (112), (222), (133), (321), (113), and (312) with an approximate diffraction angle of 23°, 29°, 37°, 47°, 51°, 56°, 58°, 64°, and 72°, respectively. This indicates that good dispersion of CuO and CuO impregnation does not do any structural damage in these supports [38]. However, the peak intensity of supports is lower-ing suggesting the impact of CuO particles on the pores of these supports [39] (Table 1). Only, Cu-COK12 catalyst shows four characteristics peaks of CuO at 35°, 38°, 54°, and 76°, indexed to (002), (111), (020), and (222) miller indices due to large CuO particles present on COK12 surface [40].

  1. No data interpretation and discussion are provided for TPR results.

Answer: Temperature-programmed reduction (TPR) is a technique that is widely used to examine metal oxide’s surface chemistry under programmed temperature conditions. TPR yields quantitative and qualitative information about the reducing characteristics of the oxide’s surface. The Micromeritics Auto Chem II apparatus was used to carry out the temperature-programmed reduction experiments. The sample was placed in the TPR cell and flushed with argon for 30 min at 150oC. The sample was then subsequently cooled down to room temperature. Finally, the furnace temperature was increased at a ramp-up rate of 10°C/min in a 15 ml/min flow rate with the H2/Ar mixture (10:90 ratios). A thermal conductivity detector (TCD) was used to monitor the signals corresponding to H2 consumption. Figure 8 represents the consumption profiles of hydrogen in the H2-TPR experiments of Cu-modified ZSM-5, Y-Zeolite, Nb2O5, and COK-12 catalysts. Expect Cu/Nb2O5 (700-900oC), all catalysts show reduction peak corresponding to CuO. Reduction of CuO include firstly CuO reduced to Cu2O then Cu2O to Cu0 and strongly in-fluence by the support, metal-support interactions, and copper dispersions [41]. Cu-COK12 and Cu-Y-Zeolite show single reduction peak indicate reduction of surface CuO. By contrast, the TPR profile of Cu-ZSM5 and Cu-Nb2O5 catalysts exhibits one additional shoulder peak suggesting reduction of surface copper oxides along with reduction of bulk copper oxide [42]. The peak shift to high temperature in Cu-ZSM5 and Cu-Y-Zeolite re-flects the strong interaction between CuO and supports [43].

  1. Why authors added Figure 9d and 9e? Why there is no peak after loading Cu over COK-12 and Nb2O5? No data interpretation and data analysis?

Answer: Thank you very much for pointing this. There is no peak after loading Cu over COK-12 and Nb2O5 The crystallite size of Cu over COK-12 and Nb2O5 might be lower than the detection level of the equipment/technique/method. 

  1. Why there is noise and different temperature ranges for NH3-TPD?

Answer: Thank you very much for your comment, the temperature range selected is of 50–700 °C. There is a noise as for Cu/Y-Zeolite, two desorption peaks were observed at 200o C and 410°C. The desorption peak at 200°C is due to physiosorbed NH3 or ammonium species, and the desorption peak at 410 °C is assigned to NH3 held on the strong acid sites. Two significant crests can be observed in the NH3-TPD profiles of Y-Zeolite, adjacent to each other, but due to Cu's addition within the location, three peaks were observed around 135, 400, 600°C temperature within the chart. The Cu/Nb2O5 catalyst shows no ammonia desorption, but in Nb2O5, there is one peak around 257-262 °C

  1. Figures 11-14, y-axis should be conversion instead of conversion rate. The x-axis units need to be same.

Answer: Thank you very much for pointing this, we have improved the figures and modified them as per your advice.

Figure 11. NO conversion rates vs temperature for various supports.

The x axis unit in figure 11-13 is temperature, but in figure 14 it is time in hours as it is the temporal distribution of NO conversion using Cu-ZSM5 catalyst.

  1. Why did author choose Cu-ZSM5 for long term stability test?

Answer: Thank you very much for your comment, this is done due to the availability of sufficient literature on reactivity tests using Cu-ZSM5. Also, Figure 11 (NO conversion rates vs temperature for various supports.) showed that ZSM5 support alone gave the highest NO conversion percentage, so it was hypothesized that the NO conversion percentage by Cu-ZSM 5 will be good enough. It proved to be good as well, when NO conversion percentage reached more than 60% at low flow rates. Although we are checking for the Cu-COK 12 also in our future work.

  1. Conclusions are required to be revised.

Answer: Thank you very much for your siuggestions the conclusions are revised and some important points are added into the new draft.

  1. Overall authors need to significantly revise and improve the content and presentation of the article in order to get it published in Solids.

Answer: Authors want to really thank you very much for your careful observation, and lots of help to improve our manuscript, we hope that all the comments are satisfactorily answered in the revised manuscript.

Round 2

Reviewer 2 Report

I reccomend to accept  and publish this manuscript.

Author Response

Thank you very much for your comments and suggestion and recommending our manuscript for publication. The required corrections are done in the final manuscript. Authors are highly obliged for all your efforts to refine the manuscript; your suggestions helped a lot in betterment of our manuscript. We are fully contended with all your comments and suggestions, it was a pleasure to hear from your side.

Reviewer 3 Report

The authors have improved the manuscript and tried to address the points raised. There are still some minor corrections required to be done before accepting the article for publication:

1- The experimental details of TPR/TPD needs to be moved to Characterization section 2.4.

2- In TPR results, degree of reduction and amount of hydrogen uptake needs to be added in tabular form.

3- The amount of NH3 adsorbed needs to be reported in tabular form.

4- The authors should compare their findings with ref 34 and discuss the outcomes

Author Response

The authors have improved the manuscript and tried to address the points raised. There are still some minor corrections required to be done before accepting the article for publication:

Answer: Thank you very much for your comments and suggestion and recommending our manuscript for publication. The required corrections are done in the final manuscript. Authors are highly obliged for all your efforts to refine the manuscript; your suggestions helped a lot in betterment of our manuscript. We are fully contended with all your comments and suggestions, it was a pleasure to hear from your side.

  1. The experimental details of TPR/TPD needs to be moved to Characterization section 2.4.

Answer: Thank you very much for your comment, the required change has been made, following details are added in section 2.4:

The Micromeritics Auto Chem II apparatus was used to carry out the temperature-programmed reduction experiments. The sample was placed in the TPR cell and flushed with argon for 30 min at 150oC. The sample was then subsequently cooled down to room temperature. Finally, the furnace temperature was increased at a ramp-up rate of 10°C/min in a 15 ml/min flow rate with the H2/Ar mixture (10:90 ratios). A thermal conductivity detector (TCD) was used to monitor the signals corresponding to H2 consumption. CO2-TPD and NH3-TPD measurements have also been completed for the prepared catalysts. TPD studies were carried out using an Autochem HP II-2950 chemisorption analyser (Micromeritics, USA). NH3-TPD was performed on Micromeritics, Autochem HP II-2950, equipped with a TCD. Before analysis, the sample was pre-treated at 300 ˚C for 1 h in He, and then it was exposed to 30 ml.min-1 flow of 10% NH3-He for 30 min. After the adsorption, it was exposed to He for 30 min to remove extra NH3-He present over the sur-face; then, the temperature was raised to 700 ˚C for TPD measurements.

  1. In TPR results, degree of reduction and amount of hydrogen uptake needs to be added in tabular form.

Answer: Authors want to thank you for your comments, we want to commit that the TPR and TPD were performed in 2018 on these catalysts, so some of the excel raw data files are missing (for 2 catalysts) in our folders as it’s been a long time. These graphs were plotted at that time, and we can promise that these are correct and reliable. We want to say that we don’t want to reattain the data from these graphs, as it may misinterpret the accuracy of the analysis. We kindly request you to please accept the graphical representations in its current form they are trustworthy and reliable. We are highly thankful for your efforts to make our manuscript better.

  1. The amount of NH3 adsorbed needs to be reported in tabular form.

Answer: Authors want to thank you for your comments, we want to commit that the TPR and TPD were performed in 2018 on these catalysts, so some of the excel raw data files are missing (for 2 catalysts) in our folders as it’s been a long time. These graphs were plotted at that time, and we can promise that these are correct and reliable. We want to say that we don’t want to reattain the data from these graphs, as it may misinterpret the accuracy of the analysis. We kindly request you to please accept the graphical representations in its current form they are trustworthy and reliable. We are highly thankful for your efforts to make our manuscript better.

  1. The authors should compare their findings with ref 34 and discuss the outcomes.

Answer: Thank you very much for your suggestion, the study performed by Curtin et al. [34] uses different weight percentages of Cu over ZSM support and our study only uses one weight percentage (3%) of Cu, so comparison would be very difficult. Although the change has been made in line number 216 and 492, stating that the NO conversion activity achieved in current study at around 3% of Cu is in coherence with what Curtin et al. [34] reported.